# Inhibitory columnar feedback neurons are involved in motion processing in *Drosophila*

**Miriam Henning[1], Madhura D Ketkar[1†], Teresa Lüffe[1], Daryl M Gohl[2,3], Thomas R Clandinin[4], Marion Silies[1,5]\***

[1]Institute of Developmental Biology and Neurobiology, Johannes-Gutenberg University Mainz, Mainz, Germany; [2]University of Minnesota Genomics Center, Minneapolis, United States; [3]Department of Genetics, Cell Biology, and Developmental Biology, University of Minnesota, Minneapolis, United States; [4]Department of Neurobiology, Stanford University, Stanford, United States; [5]Institute for Quantitative and Computational Biosciences (IQCB), Johannes Gutenberg University Mainz, Mainz, Germany

**\*For correspondence:** msilies@uni-mainz.de

**Present address:** †European Neuroscience Institute Göttingen (ENI-G), a Joint Initiative of the University Medical Center Göttingen and the Max Planck Institute for Multidisciplinary Sciences, Göttingen, Germany

**Competing interest:** The authors declare that no competing interests exist.

## eLife Assessment

This **important** article reports on the role of specific interneurons in the motion processing circuitry of the fruit fly, and marshals **convincing** evidence from neural recording, genetic manipulation, and behavioral analysis. A significant result ties the activity of C2/C3 neurons to the temporal resolution of the motion vision system. It remains unclear whether disrupting this pathway affects the dynamics of vision more generally.

**Abstract** Visual motion information is essential to guiding the movements of many animals. The establishment of direction-selective signals, a hallmark of motion detection, is considered a core neural computation and has been characterized extensively in primates, mice, and fruit flies. In flies, the circuits that produce direction-selective signals rely on feedforward visual pathways that connect peripheral visual inputs to the dendrites of the ON and OFF-direction-selective cells. Here, we describe a novel role for feedback inhibition in motion computation. Two GABAergic neurons, C2 and C3, connect to neurons upstream of the direction-selective T4 and T5 cells, and blocking C2 and C3 affects direction selectivity in T4/T5. In the ON pathway, this is likely achieved by C2-mediated suppression of responses in the major T4 input neuron Mi1. Together, C2 and C3 suppress responses to non-preferred stimuli in both T4 and T5. At the behavioral level, feedback inhibition temporally sharpens responses to ON-moving stimuli, enhancing the fly's ability to discriminate visual stimuli that occur in quick succession. GABAergic inhibitory feedback neurons thus constitute an essential component within the circuitry that computes visual motion.

## Introduction

To process sensory stimuli, information flows from the periphery to higher order brain areas, where neurons along the sensory pathway become subsequently tuned to increasingly complex stimulus features. In the visual system, photoreceptors signal intensity of light, downstream neurons sequentially extract information about ON and OFF contrast, the orientation of edges, and motion direction. Prevailing models for extracting these features, especially of motion detection in flies, rely on feedforward networks (*Currier et al., 2023*; *Ryu et al., 2022*). However, peripheral visual circuitry contains both feedforward as well as feedback connections (*Fischbach and Dittrich, 1989*; *Matsliah et al.,*

*2024*; *Nern et al., 2025*; *Meinertzhagen and O'Neil, 1991*). The functional significance of feedback neurons, particularly inhibitory feedback mechanisms, in visual motion processing is not understood.

Inhibitory feedback mechanisms are commonly found to improve the precision and accuracy of sensory processing across modalities. In the olfactory system of flies, both recurrent and reciprocal inhibition shape the processing between olfactory receptor neurons (ORNs) and local interneurons (LNs) within the antennal lobe (*Brandão et al., 2021*). Here, ORNs provide excitatory input to projection neurons (PNs) and LNs. Recurrent inhibitory feedback from LNs to ORNs enables the PNs to remain within an optimal dynamic range (*Brandão et al., 2021*; *Olsen et al., 2010*). Reciprocal inhibition from LNs to ORNs and PNs of neighboring glomeruli also enhances odor discrimination (*Olsen and Wilson, 2008*; *Wilson and Laurent, 2005*). Analogous mechanisms exist in the vertebrate olfactory bulb (*Mori et al., 1999*). In higher olfactory fly brain centers, inhibitory feedback circuits within the mushroom body improve the response selectivity of Kenyon cells to further ensure odor discriminability (*Amin et al., 2020*; *Lin et al., 2014*; *Ray et al., 2020*). Feedback inhibition also modulates the timing and intensity of auditory signals, thereby enhancing sound localization in vertebrates (*Lopez-Poveda, 2018*), and improves signal processing in the auditory system of mosquitoes (*Loh et al., 2023*). In the vertebrate retina, inhibitory feedback signals from horizontal cells and amacrine cells to photoreceptors and bipolar cells, respectively, are involved in multiple mechanisms of retinal processing including global light adaptation, spatial frequency tuning, temporal sharpening, or the center-surround organization (*Diamond, 2017*). A recent study in the fruit fly has revealed an inhibitory feedback circuit at the level of local motion detectors that enhances DS by motion opponent signaling in downstream widefield cells (*Ammer et al., 2023*). Taken together, feedback circuits improve the precision of sensory processing. Thus, similar mechanisms of feedback inhibition might also be relevant for peripheral visual processing in flies.

In the fly visual system, photoreceptors as well as lamina neurons receive synaptic input from putative feedback neurons. For example, color-opponent signaling requires reciprocal inhibition between photoreceptors as well as feedback inhibition from distal medulla (Dm) neurons (*Schnaitmann et al., 2018*; *Heath et al., 2020*; *Schnaitmann et al., 2024*). The lamina neuron L2 and the amacrine cell form a negative feedback loop to photoreceptors, which impacts their response speed and amplitude (*Fischbach and Dittrich, 1989*; *Shaw, 1984*; *Zheng et al., 2006*). The temporal response profile of lamina neurons is then sharpened by recurrent feedback (*Pang et al., 2024*). Both columnar (T1, C2, and C3) as well as wide-field neurons (lamina wide field Lawf1, Lawf2, Lat) project from the medulla back to the lamina (*Fischbach and Dittrich, 1989*; *Meinertzhagen and O'Neil, 1991*), of which only a few have been functionally characterized. The cholinergic Lawf2, which is itself modulated during active behavior by octopamine, has been proposed to subtract redundant low-frequency information from lamina neurons and might therefore allow the system to efficiently encode frequencies relevant to behavior (*Tuthill et al., 2014*). In contrast to these neurons, which synapse locally, columnar GABAergic neurons C2 and C3 make synapses throughout the entire medulla and lamina, arguing that they are ideally posed to play a broader role in regulating visual computations.

In *Drosophila*, the feedforward signals that play a central role in motion processing have been mapped in exquisite detail (*Currier et al., 2023*; *Ramos-Traslosheros et al., 2018*). Downstream of photoreceptors, lamina cells convey different types of contrast and luminance information (*Ketkar et al., 2020*; *Ketkar et al., 2022*; *Ketkar et al., 2023*) in a retinotopic fashion to the medulla. Here, medulla intrinsic (Mi) or transmedullary (Tm) neurons are either sensitive to contrast increments in the ON pathway or contrast decrements in the OFF pathway (*Behnia et al., 2014*; *Molina-Obando et al., 2019*; *Serbe et al., 2016*; *Strother et al., 2017*). They connect to the dendrites of the local ON or OFF motion detectors, T4 and T5, where information from neighboring points in visual space is integrated to compute direction-selective (DS) responses (*Fisher et al., 2015b*; *Haag et al., 2016*; *Leong et al., 2016*; *Maisak et al., 2013*; *Shinomiya et al., 2019*). The underlying feedforward circuits account well for motion responses in T4/T5, which was shown using several modeling frameworks (*Gruntman et al., 2018*; *Gruntman et al., 2019*; *Ramos-Traslosheros and Silies, 2021*; *Strother et al., 2017*; *Yang and Clandinin, 2018*; *Serbe et al., 2016*). In these core circuits for motion computation, GABAergic mechanisms are required for DS in T4/T5 (*Fisher et al., 2015b*). Interestingly, while broad pharmacological manipulation of GABAergic mechanisms in the fly visual system completely abolishes T4/T5 DS (*Fisher et al., 2015b*), a cell-type-specific loss of the major GABA receptor Rdl in

T5 only minimally reduces their DS responses (*Braun et al., 2023*). Outside the described core motion detection circuits, many cell types, including feedback neurons, are predicted to be GABAergic (*Davis et al., 2020*). However, which ones contribute to motion processing is unknown.

Here, we identified the GABAergic feedback neurons C2 and C3 as being required for early visual processing. Based on results from a behavioral forward genetic screen that suggested a role for C2 and C3 in motion-guided behavior, we characterized these neurons and their functional role in early visual processing using in vivo two-photon calcium imaging. We show that C2 and C3 are ON selective and that C2 or C3 suppress responses of T4 neurons to full-field ON stimuli and non-preferred motion directions. Blocking their synaptic output significantly reduces direction selectivity. Both neurons connect widely to lamina and medulla neurons of both ON and OFF pathways, upstream of T4/T5, arguing that the suppressive effect of C2 observed in T4 could be inherited from its major input Mi1, where inhibitory feedback from C2 sharpens the timing and reduces the gain of the response. At the behavioral level, silencing C2 causes a deficit in temporal ON edge discrimination. Taken together, feedback inhibition mediated by the two GABAergic C2 and C3 neurons is required for sharpening responses of neurons of the motion circuitry to enhance direction-selective signals in T4 and T5 cells and thus motion computation in the fly visual system.

## Results

### A forward genetic behavioral screen identifies GABAergic C2 neurons to be involved in motion detection

To identify GABAergic neurons with an essential role in early motion visual processing, we explored data from a forward genetic screen where different neuronal cell types, marked by a large collection of InSITE Gal4 driver lines, were tested for a role in motion-guided behaviors (*Gohl et al., 2011*; *Silies et al., 2013*). Because the detection of visual motion cues relies on a succession of more peripheral visual computations, this could identify neurons affecting contrast computation, temporal properties of visual neurons, but also specific aspects of motion processing.

We explored both deficits in OFF or ON motion detection. First, out of 25 driver lines that showed a deficit in OFF motion detection when synaptic activity was blocked (*Silies et al., 2013*; *Figure 1—figure supplement 1a and b*), seven lines contained GABA-positive neurons as shown by marking the Gal4-expression pattern with GFP and co-labeling brains with GABA (*Figure 1—figure supplement 1c and d*). We identified cell types within those driver lines using a Flp-out strategy to stochastically sparsify the expression pattern (*Wong et al., 2002*, *Figure 1a and b*). Out of the seven InSITE lines, expression in four lines (the PBac{IT.GAL4} lines 0787, 0564, 0301, and 0940) marked the GABAergic columnar feedback neuron C2, which is clearly recognizable based on its arborizations in the medulla (M) layer M1, M5, M8, and M10 and the distinct position of its cell body between medulla and lobula plate (*Figure 1b and e* and *Supplementary file 1*; *Fischbach and Dittrich, 1989*; *Kolodziejczyk et al., 2008*). Three of these driver lines (the PBac{IT.GAL4} lines 0787, 0564, 0940) expressed in only one or two different cell types, including C2 (*Figure 1a*), and C2 neurons were found in 70% (0787), 60% (0564, and 0301), and 100% (0940) of all Flp-out clones, indicating that in all four InSITE lines the expression pattern was dominated by C2. These results raised the possibility that C2 is an integral functional component of the visual circuitry that computes motion. An intersectional strategy between the InSITE Gal4 line and a split driver half under the promoter of *glutamic acid decarboxylase 1 (gad1-p65AD)*, a key enzyme for GABA synthesis (*Featherstone et al., 2000*), confirmed that the GABAergic C2 cell was dominantly expressed in the InSITE lines with behavioral deficits for OFF motion detection (*Figure 1c*). This strategy also revealed other GABAergic cell types, including the columnar neuron C3 and the large amacrine cell CT1, which were, however, also weakly present in the *gad1-p65AD* control (data not shown).

To next identify potential GABAergic neurons that are important for motion computation in the ON pathway, we intersected 11 InSITE-Gal4 lines with deficits in ON-edge detection (*Silies et al., 2013*) with the *Gad1* pattern (*Figure 1—figure supplement 1e and f*). This also identified three lines that consistently labeled C2, as well as other cells (*Figure 1d*, *Figure 1—figure supplement 1f* and *Supplementary file 1*). Taken together, we repeatedly found the GABAergic neuron C2 to be dominantly expressed by InSITE Gal4 lines that target cells involved in motion detection, suggesting that the GABAergic feedback neuron C2 plays an essential role in motion computation.

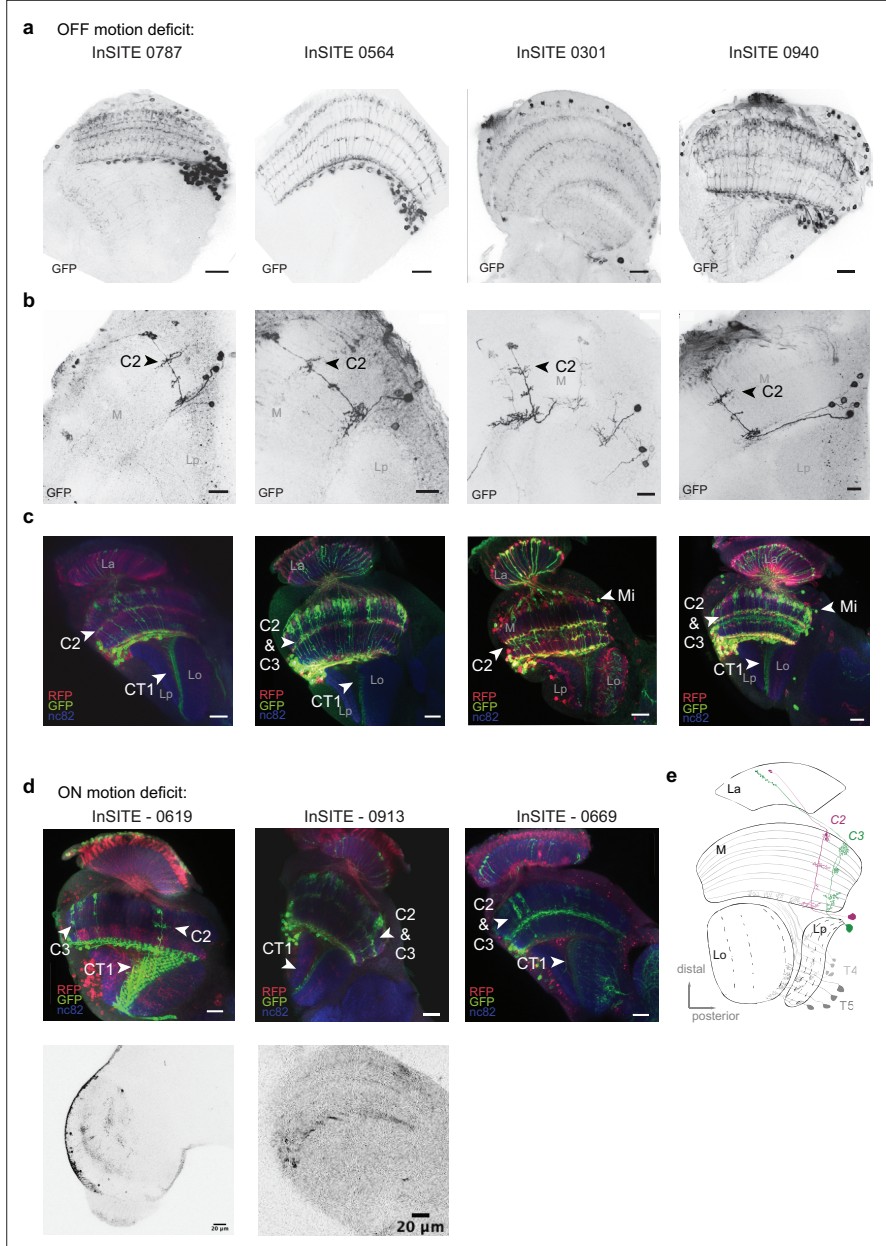

**Figure 1.** C2 dominates the expression pattern of driver lines with deficits in motion vision upon neuronal silencing. (**a–c**) Four examples of InSITE lines with behavioral deficits to OFF motion stimuli expressing the columnar feedback neuron C2. Shown are the full expression pattern (**a**, scale bar = 20 μm, 10 μm), single-cell clones (**b**, scale bar = 10 μm), and the *gad1* intersection pattern (**c**, scale bar = 20 μm), where the InSITE and *gad1* intersection (*InSITE-Gal4 UAS-LexADBD, Gad1-p65AD, lexAop-GFP*) is shown in green (GFP), the original expression pattern of the Gal4 line in red (RFP), and the neuropil is marked with nc82 (blue). The full *InSITE-Gal4* expression pattern is additionally visualized with RFP. (**d**) Three examples of InSITE lines with behavioral deficits to ON motion stimuli screened for neurons within the *gad1* intersection pattern (scale bar = 20 μm). (**e**) Drawing of the two GABAergic neurons C2 (magenta) and C3 (green) in the fly visual system, including the four neuropiles: lamina (La), medulla (M), lobula (Lo), and lobula plate (Lp). The local motion detectors T4 and T5 are additionally visualized in gray.

The online version of this article includes the following figure supplement(s) for figure 1:

**Figure supplement 1.** Screen for behaviorally relevant GABAergic neurons.

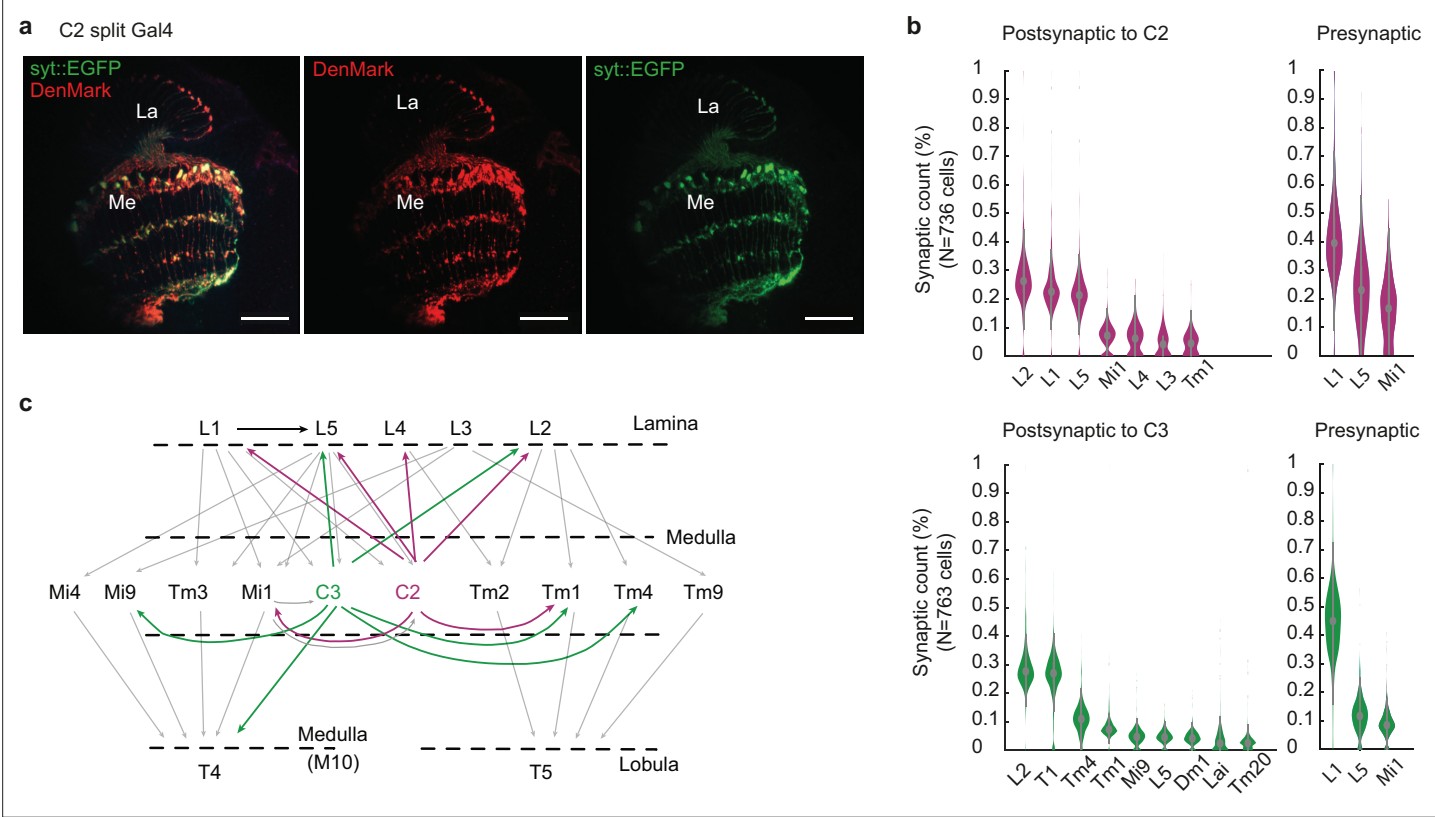

**Figure 2.** C2 and C3 connect to circuitry upstream of motion-sensitive neurons. (**a**) Confocal images of C2 cells labeled with the dendritic marker DenMark and a GFP-tagged Synaptotagmin (syt::EGFP) to label pre-synapses in C2 (scale bar = 20 μm). (**b**) Percentage of synapse counts with neurons postsynaptic and presynaptic to C2 or C3. EM data . Shown are mean ± standard error. (**c**) Illustration of the C2 and C3 circuitry within the motion detection pathway. Data in (**b–c**) were extracted from the FAFB EM dataset and Flywire connectome (*Dorkenwald et al., 2024*).

## C2 and C3 neurons synapse onto many lamina and medulla neurons of motion-detection circuits

To understand how C2 is embedded in core motion detection circuits, we next investigated the synaptic output layers and synaptic targets of C2. Using a C2 split-Gal4 line (*Triphan et al., 2016*), we expressed GFP-tagged Synaptotagmin (Syt::GFP) to label pre-synapses together with the dendritic marker DenMark (*Nicolaï et al., 2010*). C2 has both postsynaptic and presynaptic sites throughout many layers of the medulla, as well as in the distal lamina (*Figure 2a*). To understand how C2 could affect visual computation, we next explored which neurons C2 is connected to and extracted synapse counts from the FAFB EM dataset using the Flywire connectome (*Dorkenwald et al., 2024*; *Zheng et al., 2018*; *Schlegel et al., 2024*). This revealed that C2 forms several presynaptic contacts with the lamina neurons L5, L1, and L2 (*Figure 2b*). L1 and L2 constitute major inputs to the ON and OFF pathway (*Clark et al., 2011*; *Joesch et al., 2010*; *Ketkar et al., 2022*). In the medulla, C2 predominantly contacts Mi1 and Tm1, where Mi1, an ON-pathway neuron, receives its main input from L1 and provides input to T4. Tm1 plays an important role within the OFF motion pathway and provides critical input to T5 (*Serbe et al., 2016*; *Shinomiya et al., 2019*; *Takemura et al., 2013*). We extended this analysis to the C2 sibling neuron C3 because a previous study had implicated both C2 and C3 as important for motion-guided behaviors (*Tuthill et al., 2013*). C3 mainly targets T1, L2, and L5, the transmedullary neurons Tm1 and Tm4, and the medulla neuron Mi9 (*Figure 2b*). C3 also provides direct, although weak, input to the direction-selective T4 cells of the ON pathway (2.2% of T4 synapses), which was previously reported by *Shinomiya et al., 2019*. Except for T1, all these neurons are important known components of the ON and OFF motion-detection circuits (*Figure 2c*; *Currier et al., 2023*; *Ramos-Traslosheros et al., 2018*). Thus, C2 and C3 are tightly interconnected with core circuits that guide behavioral responses to motion stimuli.

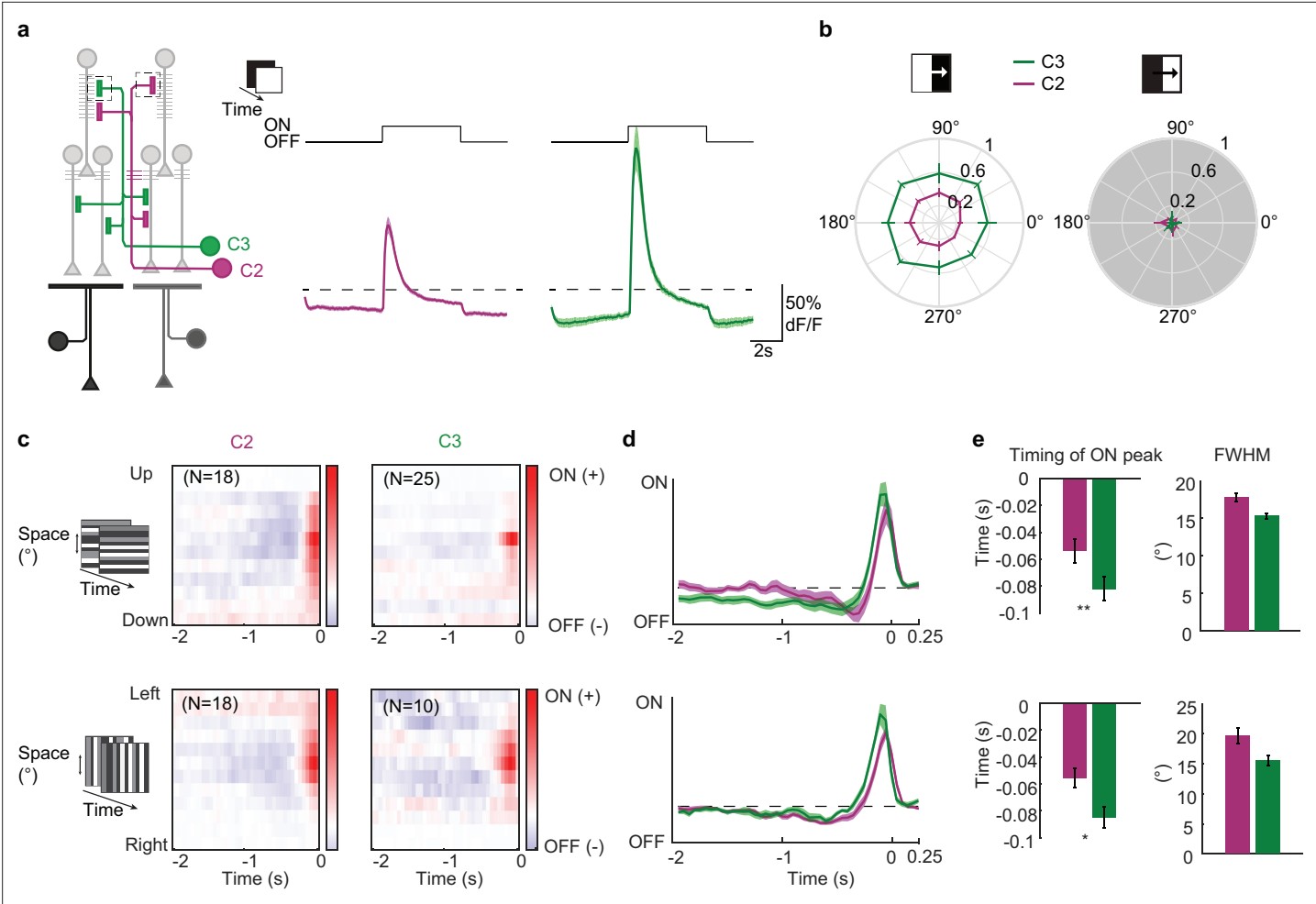

**Figure 3.** Response properties of C2 and C3 to visual stimulation. (**a**) Calcium responses of C2 (magenta, N=16 flies, 181 cells) and C3 (green, N=12 flies, 149 cells) neurons in medulla layer M1 to full field ON and OFF flashes. (**b**) Polar plots showing response amplitude of C2 (N=6 flies, 85 cells) and C3 (N=8 flies, 77 cells) to ON and OFF edges moving onto eight different directions. Shown are mean ± standard error. (**c**) Average aligned spatiotemporal receptive fields (STRFs) of C2 and C3 cells extracted in layer M1 from responses to horizontal or vertical ternary noise bars. Red and blue color indicate a positive and negative correlation with the stimulus, respectively. Sample size (**N**) equals number of cells. (**d**) Temporal filter extracted from averaging single STRFs along the time axis of the horizontal and vertical STRFs. (**e**) Timing of the ON peak of the temporal filter (left) and the full width half maximum (FWHM) of a Gaussian fit extracted along the spatial dimension of maximal response of single STRFs (right). Shown are mean ± standard error. Significances are based on Wilcoxon rank sum test (p≤0.05 *, p≤0.01**, p≤0.001***).

The online version of this article includes the following figure supplement(s) for figure 3:

**Figure supplement 1.** C2 and C3 responses are weaker in proximal medulla layers.

## C2 and C3 are ON-selective neurons that sample information across few columns

To understand how the two columnar feedback neurons contribute to the computation of motion, we characterized their response properties using in vivo two-photon calcium imaging. We expressed the calcium indicator GCaMP6f specifically in either one of the two cell types and recorded responses from arborizations in several layers of the medulla to ON and OFF light flashes. Both neurons showed a transient increase in calcium signal followed by a sustained plateau response to the onset of light and returned to baseline at light offset (*Figure 3a*). Calcium responses in layer M1 showed bigger amplitudes compared to other medulla layers, in which C2 and C3 have synaptic outputs. The kinetics of the calcium responses were similar (*Figure 3—figure supplement 1*). For all following recordings, we just focused on responses in layer M1. When flies were shown ON and OFF edges moving into eight different directions, both neurons responded to the ON

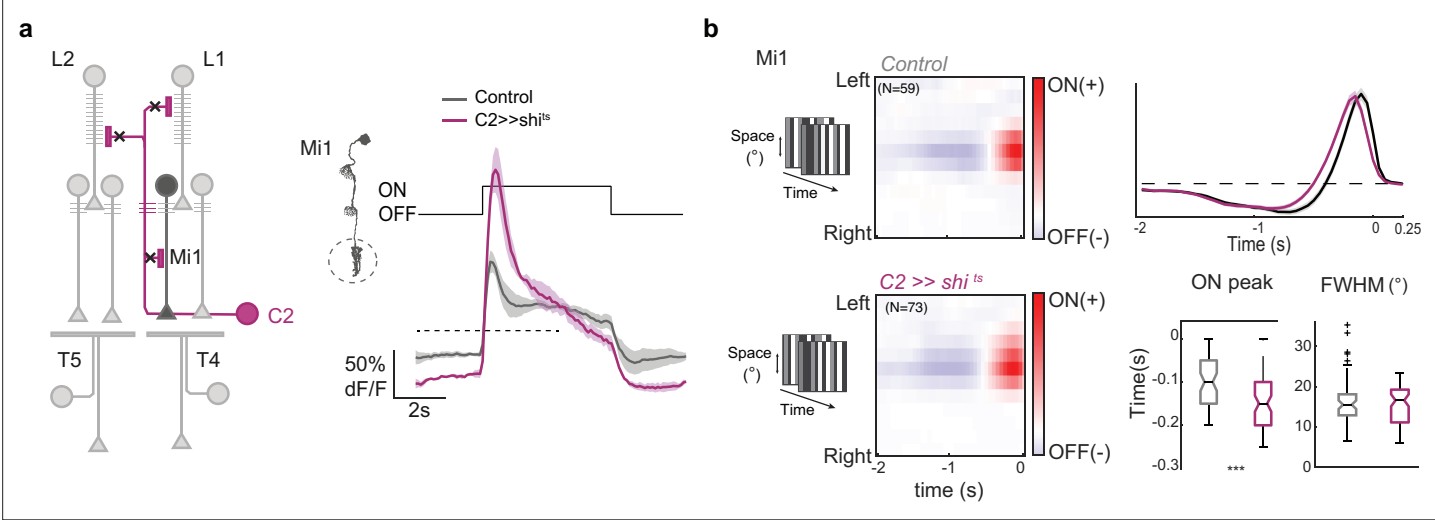

**Figure 4.** C2 shapes physiological properties of Mi1 neurons upstream of T4. (**a**) Calcium responses of Mi1 axon terminals for control condition (gray, N=5 flies, 24 cells) or while genetically silencing C2 using *shibire*[ts] (magenta, N=3 flies, 20 cells). (**b**) Left: Average aligned spatiotemporal receptive fields (STRFs) of Mi1 cells extracted from responses to horizontal or vertical ternary noise bars. Sample size (**N**) equals number of cells. Right: Temporal filter extracted from averaging single STRFs along the time axis of the horizontal and vertical STRFs. Timing of the ON peak of the temporal filter (top) and the full width half maximum (FWHM) of a Gaussian fit extracted along the spatial dimension of maximal response of single STRFs (bottom). Box plots show median, the interquartile range, and whiskers show min and max of the data. Significances are based on Wilcoxon rank sum test (p≤0.05 *, p≤0.01**, p≤0.001***).

edge but not to the OFF edge and showed no preference to any direction of motion (*Figure 3b*). To investigate the spatial and temporal filter properties of C2 and C3, we extracted their spatio-temporal receptive fields (STRFs; *Figure 3c*). Both C2 and C3 temporal filters had a fast ON and a delayed OFF response component, where the ON peak occurred significantly later in C3 than in C2 for both the vertical and the horizontal STRFs (*Figure 3c–e*). The spatial filter size, estimated by the full width half maximum (FWHM) of a Gaussian fit, ranged between ~15° and ~18° for both C2 and C3, suggesting that these neurons receive visual inputs from three columns (*Figure 3e*). Together, these data show that the GABAergic C2 and C3 neurons are functionally multicolumnar and ON selective.

## C2 neurons shape response properties of the ON pathway medulla neuron Mi1

Given the widespread connectivity of C2 and C3 in visual circuitry upstream of T4 and T5, we next asked if they shape responses of neurons from motion-detection pathways. Because C2 emerged as a prominent candidate from the behavioral screen, we focused on C2 and asked how silencing C2 affects temporal and spatial filter properties of medulla neurons that provide direct input to T4 neurons. We chose to test Mi1 as it is the medulla neuron most strongly connected to C2. We expressed the genetically encoded calcium indicator GCaMP6f specifically in Mi1 and used two-photon imaging to measure calcium signals in response to visual stimuli. C2 was silenced by expression of UAS- *shibire*[ts] (UAS-*shi*[ts]) for temporal control of the inhibition of synaptic activity. When imaging calcium responses to full field light flashes, Mi1 responded with a transient increase in calcium signal to the onset of light that decreased to a sustained plateau response (*Figure 4a*). At light offset, the calcium signal returned to baseline. C2 silencing enhanced the peak Mi1 ON responses, which then slowly decreased throughout the stimulus presentation. Spatiotemporal receptive fields of Mi1 showed a fast ON and a slow OFF subfield (*Figure 4b*), as described before (*Arenz et al., 2017*). When silencing C2 outputs, this biphasic temporal filter was significantly delayed, whereas the spatial structure of the receptive field was not altered (*Figure 4b*). Considering the many postsynaptic targets of C2 and C3, these results might generalize such that the GABAergic C2 and C3 shape properties of other lamina and medulla neurons.

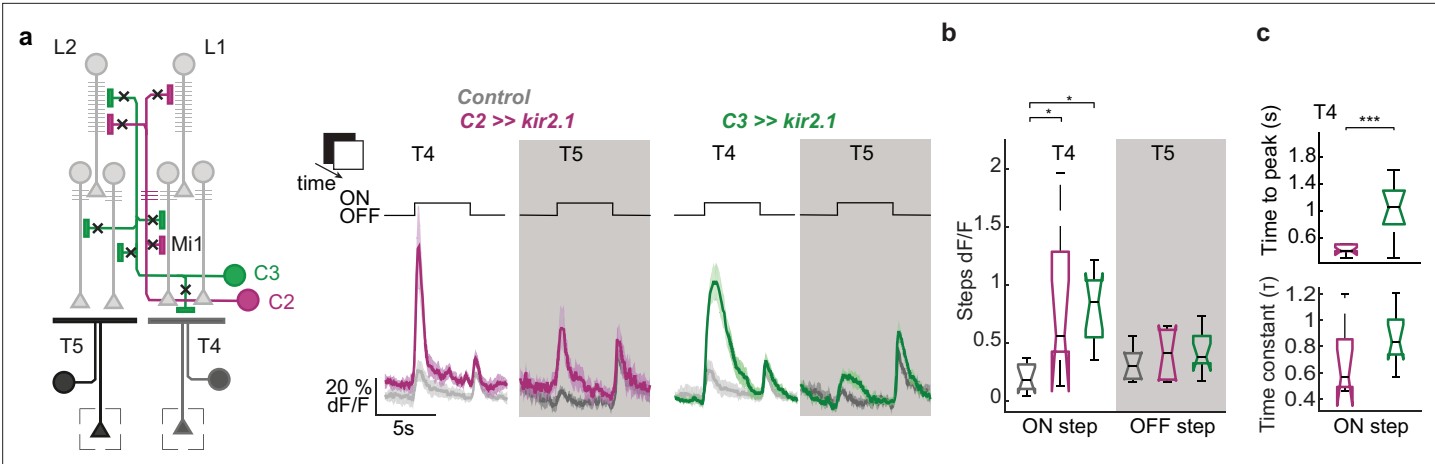

**Figure 5.** C2 and C3 disinhibit responses of motion detectors T4 and T5 to full field flash stimuli. (**a**) Schematic of visual circuitry illustrating C2 and C3 block by expressing Kir 2.1. In vivo calcium responses were recorded from axon terminals of T4 and T5 neurons (rectangles) in response to full field ON and OFF light flashes for controls (gray, N=9 flies), upon C2 block (magenta, N=8 flies) or upon C3 block (green, N=10 flies). (**b**) Calcium response of T4 neurons to the onset of light (ON step) and T5 neurons to the offset of light (OFF step) quantified from data in (**a**). (**c**) Time to peak of the ON response of T4 upon C2 and C3 block (**a**) and decay rate quantified from fitting an exponential function to the decay of the ON response. Box plots show median, the interquartile range, and whiskers show min and max of the data. Significances are based on Kruskal-Wallis Test (p≤0.05 *, p≤0.01**, p≤0.001***,+Bonferroni correction for multiple testing).

The online version of this article includes the following figure supplement(s) for figure 5:

**Figure supplement 1.** C2 and C3 suppress flash responses in T4 and T5 neurons from different lobula plate layers.

## C2 and C3 shape temporal and spatial response properties of T4 and T5 neurons

The medulla circuitry converges onto the dendrites of T4 and T5 neurons, the first direction-selective cells in the fly visual system (*Shinomiya et al., 2019*; *Takemura et al., 2013*; *Maisak et al., 2013*). To understand the compound effect of C2 and C3 on motion processing, we focused on the direction-selective T4/T5 neurons, which are downstream of many of the neurons that C2 and C3 directly connect to. We record GCaMP6f signals in T4 and T5 in response to visual stimulation. To examine if C2 and C3 affect response properties of T4/T5, we simultaneously blocked C2 and C3 by overexpression of the inward-rectifying potassium channel Kir2.1 (*Figure 5a*). Both T4 and T5 neurons hardly responded to full field ON or OFF flashes in controls (*Figure 5a*), but blocking either C2, C3, or both neurons simultaneously caused T4 neurons of all four layers to respond strongly to the onset of light (*Figure 5a and b* and *Figure 5—figure supplement 1*). While C2 block led to a fast and transient ON response in T4, the ON response caused by C3 block peaked significantly later and tended to decay slower than in the C2-blocked condition (*Figure 5a and c*).

Responses of T5 neurons to the offset of light were not affected by blocking C2 or C3 (*Figure 5b*), but T5 neurons showed a significant increase of calcium responses to the onset of light when C2 was blocked (*Figure 5a* and *Figure 5—figure supplement 1e*). This could also constitute a disinhibited ON response of T5. Together, both C2 and C3 suppress responses of the ON-selective T4 neurons to full-field flashes and reveal a possible role of C2 for suppression of ON responses in T5.

## C2 and C3 neurons sharpen direction selectivity of T4 and T5 neurons

C2 and C3 inhibit responses of T4/T5 neurons to non-motion stimuli. To test if they also impact their direction selectivity, we recorded T4/T5 responses to bars moving into eight different directions and quantified the tuning direction and the direction selectivity of T4 or T5 neurons. Blocking activity in either C2 or C3 significantly reduced direction-selectivity to ON bars in T4, and this effect was furthermore increased by blocking C2 and C3 together (*Figure 6a and b*). For OFF bars, blocking C2 but not C3 had a significant effect on direction-selectivity in T5, which was enhanced when both C2 and C3 were blocked together (*Figure 6a and b*).

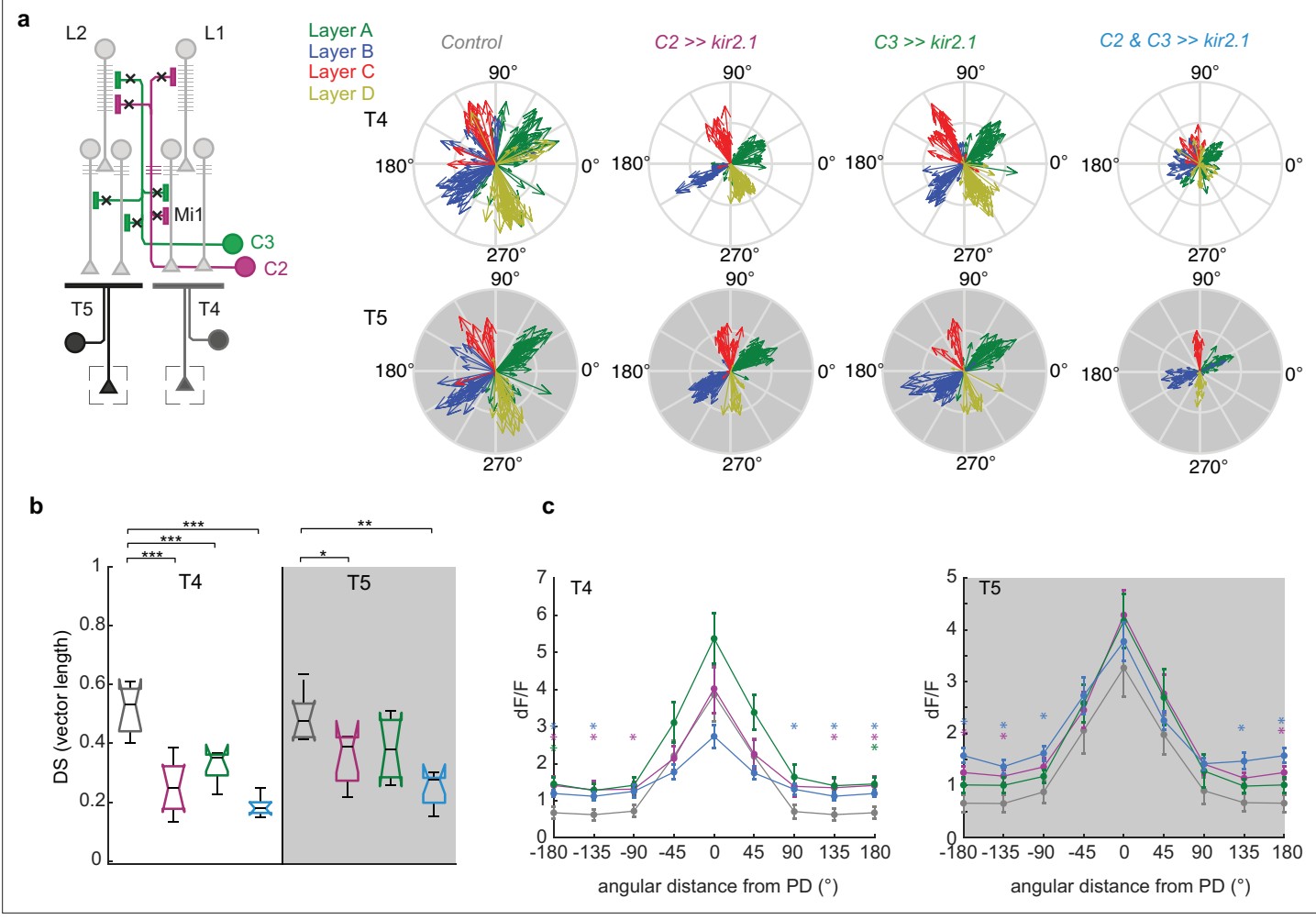

**Figure 6.** C2 and C3 are required for direction-selective responses of T4 and T5 cells. (**a**) Compass plots showing direction tuning of T4 and T5 neurons extracted from responses to ON and OFF bars moving into eight different directions for the control (*UAS-Kir2.1*), C2 block (*C2 >>Kir2.1*), C3 block (*C3 >>Kir2.1*), and C2/C3 double block (*C2&C3 >>Kir2.1*) conditions. Vector length corresponds to the strength of DS tuning. (**b**) Direction selectivity averaged across cells and flies from all layers for control condition (gray, N=7 flies, 370 T4 cells, 196 T5 cells), C2 block (magenta, N=8 flies, 170 T4 cells, 196 T5 cells), C3 block (green, N=6 flies, 188 T4 cells, 117 T5 cells), or double C2/C3 block (blue, N=7 flies, 132 T4 cells, 78 T5 cells). Box plots show median, the interquartile range, and whiskers show min and max of the data. (**c**) Response amplitude (dF/F) of T4 and T5 cells in relation to the angular distance of stimulus motion direction to the neurons PD. Shown are mean ± standard error. Significances are based on ANOVA (p≤0.05 *, p≤0.01**, p≤0.001***) (**b**) or Kruskal-Wallis test (**c**) and with Bonferroni correction for multiple testing. In (**c**) significant comparisons to the control condition are indicated by asterisks color-coded by genotype (p≤0.05 *).

The online version of this article includes the following figure supplement(s) for figure 6:

**Figure supplement 1.** C2 and C3 contribute to direction-selective responses of T4 and T5 cells from all lobula plate layers.

A loss of direction selectivity can either be due to an increase of responses to the neuron's non-preferred directions or a decrease of the response to the preferred directions (PD). To differentiate these two, we analyzed the peak responses of T4/T5 neurons to different directions of motion. To average responses of all neurons the PD of each neuron was determined by its maximal response to one of 8 directions shown. In T4, blocking C2 increased responses to motion directions that differed from the PD by more than 90°, while blocking C3 tended to enhance responses to all directions of motion, which was significant for the null direction (ND = 180°, *Figure 6c*). Simultaneously blocking the activity of C2 and C3 had a similar phenotype to C2 block alone. In T5, C2 and C3 block alone tended to slightly increase responses to all motion directions, which was significant for directions close to the null direction when C2 was blocked. Simultaneously blocking the activity of C2 and C3 enhanced these effects and significantly enhanced responses to non-preferred, or null directions in

both T4 and T5 (*Figure 6c*). Blocking C2 and C3 had similar effects for T4/T5 subtypes of every lobula plate layer (*Figure 6—figure supplement 1*). Taken together, our data show that the two columnar neurons C2 and C3 strongly impact the computation of direction selectivity, by suppressing responses to ND stimuli.

## C2 tunes the temporal dynamics of ON behavior

C2 and C3 strongly affect directional tuning of ON-direction selective cells. To test how these changes might affect behavior, we tested optomotor responses of flies walking on a spherical treadmill when silencing C2 or C3. Despite reduced T4 direction selectivity, C2 or C3 silenced flies showed a strong optomotor response within the direction of the moving stimulus (*Figure 7*, *Figure 7—figure supplement 1a–c*). However, the response dynamics during C2 silencing were different from the controls (*Figure 7a and b*). In response to a single ON edge moving onto a dark background, the turning response of controls peaked approximately 300–400ms after motion onset and declined rapidly afterwards (*Figure 7b and c*). In contrast, C2-silenced flies turned as long as the edge was moving and lacked a clear peak in their turning velocity. Deceleration of the turning response throughout the motion duration was slow, if at all present, when C2 was silenced (*Figure 7b and c*). C3-silencing had a similar but less pronounced effect on the deceleration of the turning response (*Figure 7—figure supplement 1a–c*).

We reasoned that if the C2-silenced flies are unable to discontinue their response to a stimulus, they might fail to discriminate another stimulus arriving shortly after the first one. We tested this hypothesis by presenting a pair of ON edges sequentially with individual motion duration lasting 750 ms (*Figure 7d*). Turning responses of control flies showed two distinct peaks corresponding to the two edges in the pair and recovered fully from the first response before initiating the second response (i.e. attaining a zero turning velocity; *Figure 7e*). C2-silenced flies, however, only partially recovered from the response elicited by the first edge, and the recovery was significantly less than that seen in controls (*Figure 7e and f*). This effect was even more pronounced when the individual motion duration was shortened to 500 ms, where the responses of C2-silenced flies to the two edges were now virtually indistinguishable (*Figure 7g–i*). Thus, C2 neurons are required for proper dynamics of behavioral responses to ON stimuli, particularly for the termination of these responses, and thereby for a better temporal resolution in ON motion processing. To test if there is a trade-off between temporal resolution and sensitivity in signal detection, we next asked how control and C2 silenced flies respond to the same visual stimuli at dimmer ambient light levels. Interestingly, control flies already poorly resolved the ON edges when overall light conditions were dim (*Figure 7—figure supplement 1d–o*). C2 silencing did not worsen the resolution in these dim conditions (*Figure 7j and k*, *Figure 7—figure supplement 1d–o*), suggesting a selective role of C2 in achieving better resolution in bright conditions, where the resolution does not trade-off with signal detection.

## Discussion

In this study, we identified GABAergic feedback neurons as novel components of motion computation. Both C2 and C3 feedback neurons are ON selective and connect to several lamina and medulla neurons of both ON and OFF pathways. Blocking C2 and C3 affects response properties of the local motion detector neurons, T4 and T5, and reduces direction selectivity. This is the case because blocking both C2 and C3 leads to an enhancement of ND signals in T4/T5. Given the widespread connectivity of C2 and C3 to neurons upstream of T4/T5, this effect is likely inherited from upstream neurons of T4/T5. As one example, we showed that C2 silencing leads to disinhibited responses in Mi1. The role of inhibitory feedback in the visual ON pathway also reflects at the behavioral level, where C2 silenced flies showed a behavioral deficit for discriminating moving ON edge stimuli.

### GABAergic inhibition in motion detection is mediated by C2 and C3

Several studies proposed that feedforward excitatory and inhibitory mechanisms compute direction selectivity in the T4/T5 dendrites (*Badwan et al., 2019*; *Fisher et al., 2015b*; *Haag et al., 2016*; *Leong et al., 2016*). Our data now suggest that the computation of motion signals also relies on inhibition from the two GABAergic columnar feedback neurons C2 and C3. Interestingly, C2/C3 cells and T4/T5 cells have the same developmental origin, and we now find that they are intricately linked

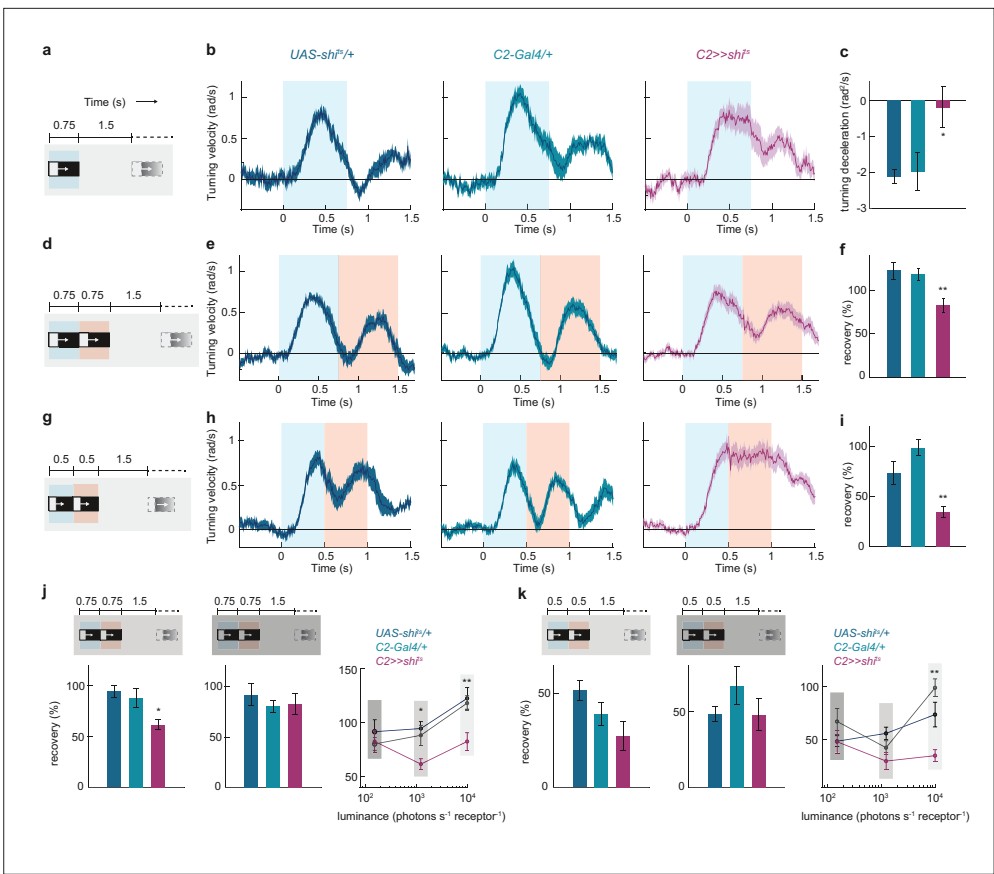

**Figure 7.** C2 is required for higher temporal resolution of the behavioral responses to moving ON edges. (**a**) Schematic of the stimulus. Single moving ON edges were presented for 0.75 s, interleaved by 1.5 s of darkness. (**b**) Time traces of control and C2-silenced flies, to the stimulus in (**a**). (**c**) Deceleration of the turning velocity in the post-peak interval (0.45 s-0.75 s) of the time traces in (**b**). *p<0.05, two-tailed Student's t tests against both controls. (**d**) A stimulus epoch comprised of two moving ON edges of 0.75 s duration each, presented one after the other without delay, and with 1.5 s darkness between two epochs. The ON edge luminance was 9806.3 photons s$^{-1}$ receptor$^{-1}$. (**e**) Time traces of control and C2-silenced flies, to the stimulus in (**d**). (**f**) Percent recovery from the turning response elicited by the first of the moving edge pair. **p<0.01, two-tailed Student's t tests against both controls. (**g**) A stimulus epoch comprised of two moving ON edges of 0.5 s duration each, presented one after the other without any delay, and with 1.5 s darkness between two epochs. (**h**) Time traces of control and C2-silenced flies, to the stimulus in (**g**). (**i**) Percent recovery from the turning response elicited by the first of the moving edge pair. **p<0.01, two-tailed Student's t tests against both controls. (**j–k**) Stimuli structured similarly to the one used in (**d**) (in j) or in (**g**) (in k) were presented at two dimmer ambient light conditions (ON edge luminance of 1225.8 and 153.2 photons s$^{-1}$ receptor$^{-1}$, respectively). The left and the middle panels show percent recovery from the turning response elicited by the first of the moving edge pair in the dimmer and the dimmest conditions. The right panel compares the recoveries across all three light conditions. *p<0.05, **p<0.01, ns: non-significant, two-tailed Student's t tests against both controls. All data show mean ± SEM. For (**b–c**), n=15 (*UAS-shi$^{ts}$/+*), n=18 (*C2-Gal4/+*), n=12 (*C2 >>shi$^{ts}$*) flies. For E-F and H-I, n=10 flies each genotype. For (j, left), n=8 (*UAS-shi$^{ts}$/+*), n=9 (*C2-Gal4/+*), n=8 (*C2 >>shi$^{ts}$*) flies. For (j, middle), n=7 (*UAS-shi$^{ts}$/+*), n=9 (*C2-Gal4/+*), n=9 (*C2 >>shi$^{ts}$*) flies. For (k, left), n=7 (*UAS-shi$^{ts}$/+*), n=9 (*C2-Gal4/+*), n=9 (*C2 >>shi$^{ts}$*) flies. For (k, middle), n=7 (*UAS-shi$^{ts}$/+*), n=9 (*C2-Gal4/+*), n=9 (*C2 >>shi$^{ts}$*) flies. Gray patches in each stimulus schematics mark the corresponding motion durations in the response-time traces in the following panel.

The online version of this article includes the following figure supplement(s) for figure 7:

**Figure supplement 1.** C2 temporally sharpens optomotor responses to moving ON edges in bright light conditions.

functionally (*Apitz and Salecker, 2016*; *Apitz and Salecker, 2018*). The involvement of C2/C3 in elementary motion detection solves a previous discrepancy between a strong loss of DS in T4/T5 in pharmacological experiments blocking GABA receptors in the whole brain (*Fisher et al., 2015b*) and weaker effects when the GABAergic inputs to T4/T5 were blocked (*Braun et al., 2023*; *Strother et al., 2017*) or when the major GABA-A-receptor Rdl was specifically knocked out in T5 (*Braun et al., 2023*). The role of GABAergic neurons for the computation of DS responses is instead likely embedded in circuitry upstream of T4/T5, where both C2 and C3 neurons are strongly connected to neurons expressing different types of ionotropic GABA$_A$ and metabotropic GABA$_B$ receptors (*Davis et al., 2020*). In contrast, there are no direct connections between C2/C3 and T4/T5 with the exception of C3 weakly synapsing onto T4 (*Shinomiya et al., 2019*). Work in blowflies has found a severe impact of GABAergic signaling for DS in LPTCs downstream of T4 and T5 cells, using application of picrotoxin to the whole brain (*Single et al., 1997*; *Schmid and Bülthoff, 1988*). Although the loss of DS in LPTCs could originate from direct inhibitory synapses onto LPTCs (*Mauss et al., 2015*; *Ammer et al., 2023*), the disruption of GABAergic signaling in upstream circuitry, which reduces DS in T4 and T5, may also contribute to the phenotype seen in LPTCs.

The physiological properties of C2 and C3 were very similar, besides the timing of their ON peak being slightly later in C3. In alignment with that, blocking C3 causes flash responses in T4 that are delayed compared to flash responses caused by C2 silencing. Furthermore, blocking C2 or C3 differentially strongly affects T4 and T5 motion responses. Interestingly, blocking C2 and C3 together has a stronger effect than blocking either one alone, supporting the idea that C2 and C3 act at least partially in parallel pathways. This is supported by the differential connectivity of C2 and C3 within core-motion detection circuitry upstream of T4/T5. Both C2 and C3 have postsynaptic partners in the lamina and in the medulla that they exclusively connect to.

Recent analysis of the full connectome of the fly visual system has shown that, among several hundred different cell types (*Matsliah et al., 2024*; *Nern et al., 2025*), only 15 cell types are truly columnar in the sense that they are present once per column and present in each column (*Nern et al., 2025*). In addition to lamina neurons, this list predominantly includes cell types with known roles in motion-computation, such as the medulla intrinsic and transmedullary neurons of core ON and OFF pathways (*Nern et al., 2025*; *Currier et al., 2023*). This list also includes both C2 and C3, which we here show to also have an important role in the establishment of direction-selective responses. Thus, our findings of the role of C2/C3 in motion computation support the interpretation that motion-detection circuits are evolutionarily constrained to have a strict columnar organization.

## How could inhibitory feedback neurons affect motion detection in the ON pathway?

C2 suppresses responses to non-motion ON flashes and ND motion in T4. While C2 does not synapse onto T4 directly, it connects to major neurons of the ON pathway, including Mi1, one of the main inputs to T4, L1, the major input to ON pathway neurons, and L5, the main input to Mi4, and can thus affect the spatial and temporal filtering properties of many T4 inputs (*Figure 2*). Indeed, C2 suppresses and temporally sharpens response kinetics of Mi1. We only tested this for this one T4 input and cannot generalize if the response properties of all neurons that are postsynaptic to C2 or C3 are affected. This might in fact not be the case, since blocking C3 alone does not affect DS responses in T5, although C3 is connected to lamina and to medulla neurons of both ON and OFF pathways. Still, the prominent connectivity of C2 and C3 to major neurons of core motion detection circuits upstream of T4/T5 and the phenotypes on T4/T5 upon C2/C3 block suggest that the effect seen is inherited from the altered response properties of at least some neurons presynaptic to T4/T5, such as Mi1 in the ON pathway.

The spatial receptive fields of C2 and C3 are consistent with the multicolumnar branching of their projections in the medulla and demonstrate that they pool information from neighboring columns. Thus, C2, with its wide receptive field structure, may also establish the inhibitory center-surround organization of several lamina and medulla neurons (*Arenz et al., 2017*; *Fisher et al., 2015a*; *Freifeld et al., 2013*; *Serbe et al., 2016*). This could be used to extract direction-selective responses, as suggested by the motion energy model, that relies on center-surround inhibition in upstream circuitry and was suggested for motion-selective cells in cat visual cortex as well as for motion detection in T4 and T5 cells (*Adelson and Bergen, 1985*; *Leong et al., 2016*). This could explain the specific increase

of ND response and T4 responses to full-field flashes when C2 is blocked. This indirect inhibition may exist alongside the suggested direct ND suppression from Mi4 and C3 to the ON pathway or CT1 in both the ON and OFF pathways (*Shinomiya et al., 2019*; *Strother et al., 2017*; *Strother et al., 2018*; *Meier and Borst, 2019*; *Takemura et al., 2017*). Such a degenerate system, employing two different mechanisms to implement the same function, may be more robust to external or internal disturbances (*Tononi et al., 1999*).

In addition to a specific role in motion computation, C2 and C3 could also more generally inhibit and balance the activity of their postsynaptic partners, which is widely discussed to ensure sparse or efficient neural coding (*Tsodyks et al., 1997*; *Avoli et al., 1995*; *Mann et al., 2009*; *Zhou and Yu, 2018*). For example, in the vertebrate retina, broadly distributed GABAergic inhibitory feedback from the amacrine cell onto bipolar cells regulates the gain or sensitivity, as well as the temporal offset of retinal ganglion cell responses (*de Vries et al., 2011*; *Nirenberg and Meister, 1997*; *Dong and Werblin, 1998*). Regulating the gain and the temporal dynamics of non-spiking neural responses, such that neuronal responses quickly return to baseline after excitation, will increase the sensitivity to subsequent events, important for temporal discrimination of stimuli. A potential role of C2 and C3 for sharpening and balancing neuronal activity in circuitry upstream of T4/T5 is in line with the observed delay and increased gain of Mi1 responses upon C2 silencing and the overall increase and less specific responses of T4 and T5 upon C2 or C3 silencing. Interestingly, C3 connects to Tm1 and Tm4, two major inputs to T5, but C3 silencing did not alter response properties in T5. However, silencing C3 together with C2 significantly enhanced phenotypes seen when C2 alone was blocked, indicating that C3 still plays a role for motion responses in T5. Whereas the role of C2 and C3 for the OFF pathway may be more generally to suppress neuronal activity, similar to the recurrent feedback connection of LNs to ORNs in the olfactory system (*Brandão et al., 2021*; *Olsen et al., 2010*), C2 seems to regulate the temporal dynamics of visual processing. Finally, the widespread connectivity of C2 and C3 to lamina and medulla neurons could also indicate that they are involved in visual computations other than motion detection, such as object detection, including the pursuit of mating partners or escape behaviors.

## Feedback inhibition is required for precision of ON motion behavior

The lack of C2 leads to more persistent behavioral responses, effectively leading to the inability to respond appropriately to two sequential visual stimuli, especially when they arrive shortly in time. Thus, inhibitory C2 feedback neurons enhance the behavioral discrimination of sequential stimulus events. Similarly, feedback inhibition in the olfactory system of flies enhances odor discrimination (*Olsen and Wilson, 2008*; *Wilson and Laurent, 2005*). These could reflect similar mechanisms to sharpen responses in either (odor) space or time (*Nagel et al., 2015*).

Visual systems increase the extent of signal summation at low light levels to improve signal-to-noise ratio and sacrifice resolution in this process (*Currea et al., 2022*; *Pick and Buchner, 1979*). In line with this, control responses poorly resolved the ON edges in dim conditions, but C2 silencing did not further worsen the temporal resolution of behavior in dim conditions, suggesting a selective role of C2 in achieving better resolution in bright conditions alone. The mechanism behind this could be simple. In dim light, lower responses of lamina and medulla neurons that in principle excite C2 will result in less feedback inhibition of C2, which does not further reduce the gain of the response but also does not sharpen the response in time. Instead, in bright light conditions, C2 feedback is strong enough to sharpen responses of the motion detection circuitry.

Previous studies have shown significant effects of both C2 and C3 silencing for other types of visually guided behaviors (*Yuan et al., 2021*; *Triphan et al., 2016*; *Tuthill et al., 2013*). For example, silencing C2 or C3 had individual effects on the amplitude and timing of behavioral responses to apparent motion stimuli with reversed contrast, so-called reverse-phi stimuli (*Tuthill et al., 2013*). Such stimuli elicit a turning response opposite to the direction of motion. Reverse-phi responses are already prominent in T4/T5 responses, likely due to the temporal decorrelation of inputs from the ON and OFF pathways (*Salazar-Gatzimas et al., 2018*; *Wienecke et al., 2018*) and will thus be affected by mechanisms that broadly shape T4/T5 response properties. C2 and C3 have also been discussed to be important for distance estimation based on parallax-motion (*Triphan et al., 2016*). It is not known how motion parallax is implemented downstream of C2 and C3 and if this involves T4/T5, but taken together, these data support the notion that C2 and C3 are an important component of visual circuitry and required for visually-guided behaviors.

# Materials and methods
## Contact for reagent and resource sharing

Further information and requests for resources and reagents should be directed to and will be fulfilled by the Lead Contact, Marion Silies (msilies@uni-mainz.de).

# Experimental model and subject details
### Key resources table

| Reagent type (species) or resource | Designation | Source or reference | Identifiers | Additional information |
|---|---|---|---|---|
| Genetic reagent (*D. melanogaster*) | *PBac{IT.GAL4}0301* | BDSC | RRID:BDSC_62767 | InSITE Gal4 screen |
| Genetic reagent (*D. melanogaster*) | *PBac{IT.GAL4}0564* | BDSC | RRID:BDSC_63411 | InSITE Gal4 screen |
| Genetic reagent (*D. melanogaster*) | *PBac{IT.GAL4}0787* | BDSC | RRID:BDSC_63782 | InSITE Gal4 screen |
| Genetic reagent (*D. melanogaster*) | *PBac{IT.GAL4}0940* | BDSC | RRID:BDSC_63911 | InSITE Gal4 screen |
| Genetic reagent (*D. melanogaster*) | *PBac{IT.GAL4}0470* | BDSC | RRID:BDSC_63341 | InSITE Gal4 screen |
| Genetic reagent (*D. melanogaster*) | *PBac{IT.GAL4}0096* | *Silies et al., 2013* | N/A | InSITE Gal4 screen |
| Genetic reagent (*D. melanogaster*) | *PBac{IT.GAL4}0396* | BDSC | RRID:BDSC_64718 | InSITE Gal4 screen |
| Genetic reagent (*D. melanogaster*) | *PBac{IT.GAL4}0619* | BDSC | RRID:BDSC_63449 | InSITE Gal4 screen |
| Genetic reagent (*D. melanogaster*) | *PBac{IT.GAL4}0913* | BDSC | RRID:BDSC_63892 | InSITE Gal4 screen |
| Genetic reagent (*D. melanogaster*) | *PBac{IT.GAL4}0669* | BDSC | RRID:BDSC_64737 | InSITE Gal4 screen |
| Genetic reagent (*D. melanogaster*) | *PBac{IT.GAL4}0974* | *Silies et al., 2013* | N/A | InSITE Gal4 screen |
| Genetic reagent (*D. melanogaster*) | *PBac{IT.GAL4}1037* | BDSC | RRID:BDSC_63975 | InSITE Gal4 screen |
| Genetic reagent (*D. melanogaster*) | *PBac{IT.GAL4}0980* | BDSC | RRID:BDSC_63936 | InSITE Gal4 screen |
| Genetic reagent (*D. melanogaster*) | *PBac{IT.GAL4}0518* | *Silies et al., 2013* | N/A | InSITE Gal4 screen |
| Genetic reagent (*D. melanogaster*) | *PBac{IT.GAL4}0756* | BDSC | RRID:BDSC_63499 | InSITE Gal4 screen |
| Genetic reagent (*D. melanogaster*) | *PBac{IT.GAL4}0081* | BDSC | RRID:BDSC_62703 | InSITE Gal4 screen |
| Genetic reagent (*D. melanogaster*) | *PBac{IT.GAL4}0168* | BDSC | RRID:BDSC_62706 | InSITE Gal4 screen |
| Genetic reagent (*D. melanogaster*) | *PBac{IT.GAL4}0651* | BDSC | RRID:BDSC_63731 | InSITE Gal4 screen |
| Genetic reagent (*D. melanogaster*) | *UAS-FRT-CD2y+-RFT-mCD8::GFP;* | *Wong et al., 2002* | N/A | *UASFlp* FlpOut clones |
| Genetic reagent (*D. melanogaster*) | *UAS-CD8::GFP(I); UAS-2xEGFP(II)* | BDSC | N/A | InSITE Gal4 screen |

*Continued on next page*

*Continued*

| Reagent type (species) or resource | Designation | Source or reference | Identifiers | Additional information |
|---|---|---|---|---|
| Genetic reagent (*D. melanogaster*) | *UAS-DenMark, UAS-syt.eGFP* | BDSC | RRID:BDSC_33065 | pre- and post-synaptic markers |
| Genetic reagent (*D. melanogaster*) | *UAS-mCD8::RFP$^{attP8}$; lexAop-mCD8::GFP$^{attP16}$* | BDSC | RRID:BDSC_32229 | *Gad1* intersection |
| Genetic reagent (*D. melanogaster*) | *UAS-LexA.DBD* | BDSC | RRID:BDSC_56528 | *Gad1* intersection |
| Genetic reagent (*D. melanogaster*) | *Gad1$^{MI09277}$-p65AD* | BDSC | RRID:BDSC_60322 | *Gad1* intersection |
| Genetic reagent (*D. melanogaster*) | *lexAop-GCaMP6f-p10$^{su(Hw)attP5}$* | BDSC | RRID:BDSC_44277 | Calcium Imaging |
| Genetic reagent (*D. melanogaster*) | *20xUAS-IVS- GCaMP6f $^{attP40}$* | BDSC | RRID:BDSC_42747 | C2/C3 imaging |
| Genetic reagent (*D. melanogaster*) | *R59E08-LexA$^{attP40}$* | BDSC | RRID:BDSC_52832 | T4/T5 imaging |
| Genetic reagent (*D. melanogaster*) | *R20C11-p65.AD$^{attP40}$* | BDSC | RRID:BDSC_70106 | C2 split Gal4 C2/C3 split Gal4 |
| Genetic reagent (*D. melanogaster*) | *R25B02-Gal4.DBD$^{attP2}$* | BDSC | RRID:BDSC_68969 | C2 split Gal4 |
| Genetic reagent (*D. melanogaster*) | *R26H02-p65.AD$^{attP40}$* | ***Tuthill et al., 2013*** | RRID:BDSC_70159 | C3 split Gal4 |
| Genetic reagent (*D. melanogaster*) | *R29G11-Gal4.DBD$^{attP2}$* | ***Tuthill et al., 2013*** | N/A | C3 split Gal4 |
| Genetic reagent (*D. melanogaster*) | *R48D11-Gal4.DBD$^{attP2}$* | BDSC | RRID:BDSC_69028 | C2/C3 split Gal4 |
| Genetic reagent (*D. melanogaster*) | *R19F01-lexA$^{attp40}$* | BDSC | RRID:BDSC_52547 | Mi1 lexA |
| Genetic reagent (*D. melanogaster*) | *UAS-KCNJ2.EGFP(Kir2.1)$^7$* | BDSC | RRID:BDSC_6595 | C2/C3 block |
| Genetic reagent (*D. melanogaster*) | *UAS-shi$^{ts}$* | BDSC | RRID:BDSC_44222 | C2/C3 block |
| Software, algorithm | ImageJ | National Institutes of Health | http://imagej.nih.gov/ij | |
| Software, algorithm | Imaris | Oxford Instruments | | |
| Software, algorithm | Adobe Photoshop 2021 | | | |
| Software, algorithm | MATLAB | Mathworks | The MathWorks Inc50 Natick, MA | |
| Software, algorithm | Python 2.7 | Python | https://python.org | |
| Antibody | Polyclonal anti-GFP (chicken) | Abcam | Ab13970 | Conc (1:2000) |
| Antibody | Monoclonal anti-Bruchpilot (nc82) (mouse) | DSHB | N/A | Conc (1:25) |
| Antibody | Polyclonal anti-GABA (rabbit) | Sigma-Aldrich | A2052 Sigma | Conc (1:200) |
| Antibody | Polyclonal anti-DsRed (rabbit) | Clontech | 632475 | Conc (1:400) |
| Antibody | Alexa Fluor anti-chicken-Alexa 488 (goat) | Dianova | 103-545-155 | Conc (1:200) |
| Antibody | Alexa Fluor anti-mouse-Alexa 647 (goat) | DSHB | 115-605-003 | Conc (1:200) |
| Antibody | Alexa Fluor anti-rabbit-Alexa 594 (goat) | Sigma-Aldrich | 111-585-003 | Conc (1:200) |

*Drosophila melanogaster* were raised on molasses-based food at 25 °C and 55% humidity in a 12:12 hr light-dark cycle. For all experiments, female flies were used. For imaging experiments, flies were recorded 3–5 days after eclosion at room temperature (RT, 20 °C). Genotypes used in all experiments are given in the key resources table.

## Immunohistochemistry and confocal microscopy

Fly brains were dissected and simultaneously fixed in 2% paraformaldehyde in phosphate-buffered lysine (PBL) for 1 hr at RT. Next, brains were washed 3 x in phosphate-buffered saline containing 0.3% Triton X-100 (PBT, pH 7.2) and blocked for 30 min in 10% normal goat serum (NGS, Fisher Scientific GmbH, Schwerte, Germany) in PBT at RT. The primary antibody solution was incubated for 24 hr at 4 °C and removed by washing 3 x for 5 min in PBT. Subsequently, the secondary antibody solution was incubated at 4 °C overnight. The primary and secondary antibodies used for the different experiments are listed in the key resources table. Last, the samples were washed with PBT 3 x and embedded in Vectashield (Vector Laboratories, Burlingame) for up to 7 days before imaging.

For confocal microscopy, brains were mounted with a small drop of Vectashield on a microscope slide and covered with a cover slip (Thermo Fisher Scientific GmbH, Schwerte, Germany). Serial Z-stacks were taken on a Zeiss LSM10 microscope (Carl Zeiss Microscopy GmbH, Germany) using the Zen 2 Blue Edition software (Carl Zeiss Microscopy, LLC, United States). Z-section images were taken 1 µm steps apart and at 512x512 pixel resolution using the Plan-Apochromat objective DIX(UV)VIS-IR 40 x/1.3 M27 (oil). Confocal stacks were further rendered into two-dimensional images using *Imaris* (Oxford Instruments) and later edited using *Adobe Photoshop 2021*.

## Anatomy

The expression pattern of 25 InSITE-Gal4 lines that were previously identified in a behavioral screen for motion detection deficits and that showed extensive expression in the visual system (*Silies et al., 2013*) was visualized with *UAS-eGFP* and *UAS-mCD8::GFP*, and stained with anti-GFP and anti-GABA antibodies (key resources table). A Flp-out strategy, expressing *UAS >CD2,y⁺>mCD8::GFP* as well as *UAS-Flp* identified individual cell types in the expression pattern of the InSITE Gal4 lines.

To intersect the InSITE Gal4 expression pattern with cells expressing *glutamic acid decarboxylase 1 (gad1)*, a split-LexA DNA binding domain was expressed within the InSITE Gal4 pattern using *UAS-LexA.DBD*. The p65 activation domain was expressed under the control of the gad1 promoter (*gad1-p65.AD*). Cell types present in both expression patterns thus expressed GFP when crossed to *lexAop-mCD8::GFP*. The complete InSITE expression pattern was additionally marked with *UAS-mCD8::RFP*.

To identify the dendrites and synaptic axons of C2, we expressed the dendritic marker DenMark and the GFP-tagged Syt1 as a presynaptic marker.

## Two-photon calcium imaging

Prior to two-photon imaging, flies were anesthetized on ice and fit into a small hole in stainless-steel foil, located in a custom-made holder. The head was tilted approximately 30° to expose the back of the head. A small drop of UV-sensitive glue (Bondic) was applied to the left side of the head and the thorax to fix the head of the fly. The cuticle on the right backside of the head, fat bodies, and tracheae were removed using breakable razor blades and forceps. The brain was perfused with a carboxygenated saline containing 103 mM NaCl, 3 mM KCl, 5 mM TES, 1 mM NaH₂PO₄, 4 mM MgCl₂, 1.5 mM CaCl₂, 10 mM trehalose, 10 mM glucose, 7 mM sucrose, and 26 mM NaHCO₃ (pH ~7.3). To record calcium activity, a two-photon microscope (Bruker Investigator, Bruker, Madison, WI, USA), equipped with a 25 x/1.1 objective (Nikon, Minato, Japan) was used. For excitation of GCaMP6f, a Spectraphysics Insight DS +excitation laser was tuned to a wavelength of 920 nm with <20 mW of laser power measured at the objective. Emitted light was filtered through an SP680 short pass filter, a 560 lpxr dichroic filter, and a 525/70 emission filter and detected by a PMT set to a gain of 855 V. Imaging frames were acquired at a frame rate of ~15–20 Hz and 4–7 optical zoom using the PrairieView software. Simultaneous silencing of synaptic transmission using shibire^ts was achieved upon pre-incubation in a 37° water bath for 1 hr.

## Visual stimulation

Visual stimuli were presented on a 9 cm-by-9 cm rear projection screen positioned in front of the fly and covering a visual angle of ~80° in azimuth and ~55° in elevation. Stimuli were filtered through a 482/18 bandpass filter (Semrock) and ND1.0 neutral density filter (Thorlabs) and projected using a LightCrafter 4500 DLP (Texas Instruments, Texas, USA) with a frame rate of 100 Hz and synchronized with the recording of the microscope as described previously (*Freifeld et al., 2013*).

Visual stimuli were generated using custom-written software using C++ and OpenGL. T4/T5 recordings always started by showing ON and OFF edges moving into four directions of motion used for the automated ROI selection. Presentation of subsequent stimuli was pseudo-randomized. If the fly moved, the moving edge stimulus was repeated in between recordings.

### Moving OFF and ON edges

Full contrast dark or bright edges moved across the screen with a velocity of 20°/s. Edges moving in four-directions were used for the automated identification of T4 and T5 axon terminals. An eight-direction stimulus was shown to quantify direction-selectivity of C2 and C3. Each stimulus direction was presented at least two times in pseudorandom sequence.

### Periodic full-field flashes

The full-field flash stimulus consisted of full-contrast ON and OFF flashes covering the whole screen. Each flash lasted for 5 s and thus 10 s for one stimulus epoch, presented for ~2 min per fly. This stimulus was shown to flies while recording C2/C3, T4/T5, or Mi1 cells.

### Moving OFF and ON bars

Full contrast dark or bright bars of 5° width moving with a velocity of 20°/s in eight different directions in pseudo-randomized order. This stimulus was repeated at least three times per fly and was used to quantify tuning preferences of T4 and T5 neurons.

### Ternary white noise

Ternary white noise was used to extract the spatiotemporal receptive fields (STRFs) of Mi1 and C2/C3 neurons. Each stimulus frame consisted of 12 bars of 5° x 60° size tilted along either azimuth or elevation. Each bar changed its contrast at 20 Hz with equal probability of having either minimal, maximal, or intermediate contrast independent of all other bars. The sequence ran for 500 s. The stimulus was normalized to have values of –1, 0, and 1 for dark, gray, and bright bars.

## Behavioral experiments

2–5 days old female flies were sedated using cold anesthesia and glued to the tip of a needle at the dorsal thorax, using a UV-hardened adhesive (Norland optical adhesive). The fly was positioned above a polyurethane ball (Kugel-Winnie, Bamberg, Germany), 6 mm in diameter. The ball floated on air at the center of a semi-cylindrical LED arena (*Reiser and Dickinson, 2008*). The LED panels arena (IO Rodeo, CA, USA) comprised 570 nm LEDs that spanned 192° in azimuth and 80° in elevation. The pixel resolution was 2° at the fly's elevation. The arena was housed in a dark chamber maintained at 34 °C, the restrictive temperature for *shibire*[ts]. Rotation of the ball was sampled at 120 Hz with two wireless optical sensors (Logitech Anywhere MX 1, Lausanne, Switzerland), directed towards the center of the ball and positioned at 90° to each other (described in *Seelig et al., 2010*). MATLAB coordinated stimulus presentation and data acquisition; mouse data acquisition was mediated by C#.

## Visual stimulation for behavior

The LEDs can show 16 different, linearly spaced, intensity levels. The luminance of each of these levels was measured at the fly's position using a LS-100 luminance meter (Konica Minolta, NJ, USA). These values, originally recorded in candela/m², were converted to photons incident per photoreceptor per second (photons s⁻¹ receptor⁻¹), following the procedure described by *Dubs, 1982*.

Flies were tested in an open-loop paradigm. Each fly recording comprised 65–70 stimulus epochs for the single-edge experiments, and 80–100 epochs for paired-edge experiments. The epochs were interleaved by a 1.5 s dark screen. Moving ON edges were of 100% contrast. ON edge luminance of the brightest ON edge was 0.43 cd/m$^2$ or 9806.3 photons s$^{-1}$ receptor$^{-1}$, whereas the two dimmer ON edges tested corresponded to 1225.8 and 153.2 photons s$^{-1}$ receptor$^{-1}$. Stimuli were presented in mirror-symmetric fashion (i.e. clockwise or rightward and anti-clockwise or leftward) to account for potential biases. The data were averaged after inverting the sign of the responses to leftward motion, which were originally marked as negative.

In experiments in which a single ON edge was shown, each epoch comprised a single ON edge that moved at 192°/s for 0.75 s. In paired-edge experiments, each epoch comprised two sequentially presented ON edges (one at a time but without a delay in between) that moved at 240°/s. Between single edges presentation, the screen turned black. In two variations of this paradigm, each individual edge moved for 0.75 s and thus the motion epoch lasted for 1.5 s, or each individual edge moved for only 0.5 s, and the motion epoch lasted for 1 s.

## EM data analysis

Pre- and post-synaptic counts of C2 and C3 neurons were extracted from the EM dataset of a full adult female brain (FAFB) (*Zheng et al., 2018*) using Flywire (*Dorkenwald et al., 2024*; *Schlegel et al., 2024*). Connections that were not proofread and indicated with a nan in the dataset were not considered. Similar to previous work from our lab, we only considered connections with ≥3 synapses (*Cornean et al., 2024*). To focus on the main synaptic partners, we displayed only neuron types that were connected to at least 70% of all C2 or C3 neurons of the dataset.

## Processing of two photon data

### Preprocessing

All data analysis was performed using MATLAB R2017a (The MathWorks Inc, Natick, MA) or Python 2.7. Motion artifacts were corrected using Sequential Image Alignment SIMA, applying an extended Hidden Markov Model (*Kaifosh et al., 2014*).

### ROI selection

For analysis of C2 and C3 as well as Mi1 imaging data, the average intensity of the time series was used to guide the manual selection of axon terminals using a custom written user interface in MATLAB. Responses of pixels belonging to one ROI were averaged and saved for further analysis.

To distinguish responses of single T4 and T5 axon terminals, we automatically extracted ROIs based on their unique contrast- and direction-selective responses to ON and OFF edges moving into four directions. First, the aligned images were averaged across time, and the average image intensity was Gaussian filtered (s=1.5) and then threshold-selected by Otsu's method (*Otsu, 1979*) to find foreground pixels suitable for further analysis. After averaging responses across stimulus repetitions, we selected pixels that showed a peak response larger than the average response plus two (three, for STRFs) times the standard deviation of the full trace. These pixels were grouped based on their contrast preference (ON or OFF pixels) and further assigned to four categories based on their anatomical location within the lobula plate (layers A, B, C, or D).

The preferred direction (PD) was first determined as the direction that elicited the highest response, and the null direction (ND) to the direction that was 180° offset to the PD. Similarly, the preferred contrast (PC) was set to the contrast that elicited the highest response, and the non-preferred contrast (NC) to the opposing contrast. Peak dF/F values were then used to calculate a direction-selectivity index (DSI) and contrast selectivity index (CSI) for each pixel as follows:

$$DSI = \frac{PD - ND}{PD},$$

$$CSI = \frac{PC - NC}{PC}.$$

We excluded all pixels that did not exceed the CSI threshold of 0.5 for STRF calculation to obtain clean T4 or T5 responses. For final clustering, the Euclidean distance between each pair of pixels was

calculated for the x and y location of the pixel and the timing of the PD response, and average-linkage agglomerative hierarchical clustering was performed. We further determined the optimal distance threshold that yielded most clusters between 1 and 6.5 µm², an appropriate size for T4 or T5 axon terminals. All clusters that fell outside this range were excluded from analysis. Cluster locations were saved and matched with subsequent recordings of the same cells to other stimuli.

### Response quantification

After subtracting a background signal, the response of a cell was calculated as dF/F as:

$$\frac{dF}{F} = \frac{F - F_0}{F_0},$$

where $F_0$ denotes the baseline fluorescence. Responses of single ROIs were averaged for each fly and interpolated at 10 Hz before averaging across flies.

### Full-field flashes

$F_0$ were either computed from averaging responses to a non-stimulus epoch prior to the start of stimulation (for T4/T5 data, **Figure 5**), or as the average of the whole response trace (for C2/C3 data, **Figure 3**). Responses to ON or OFF steps were quantified as the difference in dF/F averaged across a 2 s time window before the onset or offset of light and the peak response in an epoch 2 s after the onset or offset of light, respectively. The time to peak was quantified as the time delay from the onset of light to the response maximum. The decay constant was extracted from a single term exponential fit to the response 3 s after the onset of light. A decay constant was only taken into account if it was negative to filter out noisy responses.

### Moving OFF and ON bars

$F_0$ for dF/F was calculated as described above for the full-field flash stimulus. To quantify direction selectivity (DS) of single units, responses were trial averaged, and the peak response to the eight different directions of either ON or OFF bars was extracted for T4 and T5 cells, respectively. We quantified the tuning of single cells by computing vector spaces after (**Mazurek et al., 2014**):

$$L_{dir} = \left| \frac{\sum_k R\left(\theta_k\right) \exp\left(i\theta_k\right)}{\sum_k R\left(\theta_k\right)} \right|.$$

where $R\left(\theta_k\right)$ is the response to angle $\theta_k$. The direction of the vector $L_{dir}$ denotes the tuning angle of the cell, and the normalized length of the vector is related to the circular variance and thus represents the directional selectivity of the cell.

### Space-time receptive field mapping

Space-time receptive fields (STRFs) were extracted from responses of single cell ROIs to the ternary white noise stimulus. The raw fluorescence (F) traces of single clusters were extrapolated to 20 Hz matching the update rate of the ternary white noise stimulus. The response changes of the cell $\left(dF/F\right)$ were extracted as described above for the full-field flash stimulus. Here $\left(F_0\right)$ denotes the baseline fluorescence, computed from averaging responses to gray interleaves. The extracted cell response was further centered around its mean and averaged across two stimulus epochs if it was repeated twice. The stimulus was normalized to have values of –1, 0, and 1 for dark, gray, and bright bars.

STRFs were extracted by computing a weighted stimulus average also known as reverse correlation. For this, a sliding average of 2 s length was propagated backwards in time and weighted by the response of the cell at the start of the window. Given the response of the cell at time point t $\left(r_t\right)$, the time window of the stimulus $\left(\tau\right)$, the amount of total time points (T), and the stimulus snippet $s\left(t - \tau\right)$, the STRF was computed as follows:

$$STRF = \frac{1}{T - \tau} \sum_{t=\tau}^{T} r_t s\left(t - \tau\right).$$

To evaluate how well STRFs predicted the cell response, we convolved the extracted STRFs with the stimulus. This prediction of the cell response was then correlated with the measured cell response. The correlation value $R^2$ was used to discard non-valid STRFs. The $R^2$ threshold of 0.26 was chosen upon visual inspection of STRFs.

The weighted average STRF was computed by averaging all STRFs of one subtype of cell and weighting each STRF by the $R^2$ value. Therefore, STRFs that well predicted the cell response had a higher impact on the average. For visualization of each weighted average STRF, the color bar was chosen to represent the highest absolute value with the darkest color (red for highest positive, or blue for highest negative).

The temporal filter was extracted from averaging single STRFs along the time axis of the horizontal or vertical STRFs. Timing of the ON peak was calculated as the time of highest correlation of the temporal filter, and the full width half maximum (FWHM) was extracted from a Gaussian fit along the spatial dimension of maximal response of single STRFs.

## Processing of behavioral data

Behavioral responses of flies to moving stimuli were analyzed using MATLAB. Yaw velocities of the flies were derived from the optical sensors' data, as in *Seelig et al., 2010*. Velocities in the direction of stimulus motion were considered positive, and those against the stimulus, negative. Responses to each mirror-symmetric stimulus pair were aggregated when computing the trial-averaged response of a fly. Time series of turning responses presented here consist of velocities averaged across flies ± SEM. Flies with a forward walking speed less than 3 mm/s, as well as flies turning in the opposite direction of the stimulus (potentially due to dominant phototaxis), were discarded from the analysis. Together, this resulted in rejection of approximately 25% of all flies.

For the single-edge experiment, the time window 0.45 s-0.75 s with respect to motion onset was considered to analyze as the declining phase of the response. The slopes of a linear fit to the trial-averaged response of each fly were analyzed statistically.

For the paired-edge experiments, percent recovery from the response elicited by the first edge was quantified. For this purpose, two measures of velocities were extracted for each fly: peak turning velocities $V_{peak}$ were extracted from the mean of the eight highest trial-averaged instantaneous yaw velocities over the first of the paired edge durations (0–0.75 s or 0.75–1.5 s), and lowest turning velocities $V_{lowest}$ were extracted from the mean of the eight lowest instantaneous yaw velocities in the early second-edge durations (0.5–0.75 s or 0.75–1 s). The difference between these two velocities represented the recovery from the first edge of the pair. The percent recovery was calculated as

$$\% \, recovery = \frac{V_{peak} - V_{lowest}}{V_{peak}} * 100$$

To examine statistical differences between genotypes, a two-tailed Student's t test was used. Data points were considered significantly different only when the experimental group differed from both genetic controls. Data points were compared only within each luminance condition, that is the experimental group measurement in the dimmest condition was compared with the two genetic controls measured in the dimmest condition only. Note that the % recovery was first calculated per fly, that is considering $V_{peak}$ and $V_{lowest}$ specific to that fly, before the mean ± SEM statistics of these recoveries were plotted. Across flies, the $V_{peak}$ and $V_{lowest}$ occur at different time points (e.g. for *C2>>shi^{ts}* flies) and thus may not align to the same point in the fly-averaged velocity traces.

## Statistics

All statistics were done in MATLAB. For comparisons between C2 and C3 response characteristics (*Figure 2d and e*), a two-sided Wilcoxon signed rank test was applied.

For multiple comparisons of T4 and T5 response characteristics from control and C2/C3 block conditions, all samples were first tested for normal distribution with a Kolmogorov–Smirnov test. If all samples were normally distributed, a one-way ANOVA with subsequent pairwise comparison was used. Bonferroni correction was applied for multiple comparisons. In the case of non-normally distributed data, a Kruskal-Wallis test was used. We show significant differences in figures using asterisks

(p≤0.05 *, p≤0.01**, p≤0.001***). Non-significant differences are not further indicated. Statistical summaries are listed in *Appendix 1—Tables 1–7*.

## Acknowledgements

We are grateful to Christine Gündner and Jonas Chojetzki for excellent technical support and to all members of the Silies lab for valuable discussions. We thank Carlotta Martelli and Christopher Schnaitmann for critical comments on the manuscript, and Maria Ioannidou for EM data analysis. This project has received funding from the European Research Council (ERC) under the European Union's Horizon Europe research and innovation program (Grant agreement No. 101045003 'Adaptive Vision') and from the German Research Foundation (DFG) through the collaborative research center 1080 "Neural homeostasis" (project C06) to MS.

## Additional information

### Funding

| Funder | Grant reference number | Author |
|---|---|---|
| European Research Council | 101045003 | Marion Silies |
| Deutsche Forschungsgemeinschaft | CRC1080 project C06 | Marion Silies |

The funders had no role in study design, data collection and interpretation, or the decision to submit the work for publication.

### Author contributions

Miriam Henning, Conceptualization, Formal analysis, Investigation, Visualization, Writing – original draft, Writing – review and editing; Madhura D Ketkar, Formal analysis, Investigation, Visualization, Writing – original draft, Writing – review and editing; Teresa Lüffe, Investigation, Writing – review and editing; Daryl M Gohl, Formal analysis, Methodology, Writing – review and editing; Thomas R Clandinin, Conceptualization, Writing – review and editing; Marion Silies, Conceptualization, Supervision, Funding acquisition, Methodology, Writing – original draft, Project administration, Writing – review and editing

### Author ORCIDs

Miriam Henning ⓘ https://orcid.org/0000-0002-8816-7408
Madhura D Ketkar ⓘ https://orcid.org/0000-0002-0465-5616
Daryl M Gohl ⓘ https://orcid.org/0000-0002-4434-2788
Thomas R Clandinin ⓘ https://orcid.org/0000-0001-6277-6849
Marion Silies ⓘ https://orcid.org/0000-0003-2810-9828

Reviewer #1 (Public review): https://doi.org/10.7554/eLife.108529.3.sa1
Reviewer #2 (Public review): https://doi.org/10.7554/eLife.108529.3.sa2
Author response https://doi.org/10.7554/eLife.108529.3.sa3

## Additional files

### Supplementary files

Supplementary file 1. List of behaviorally relevant neurons identified from the expression pattern of InSITE lines with behavioral deficits to either OFF- or ON edge motion stimuli. Neurons were identified based on either colocalization of the InSITE expression pattern with a GABA antibody, followed by single cell Flp-Out experiments, or a InSITE-Gal4-Gad1-intersection strategy.

MDAR checklist

## Data availability

In vivo calcium imaging data produced for this study, as well as behavioral datasets, and the relevant analysis code are available at https://github.com/silieslab/C2C3_feedback_Henning (copy archived at *Silieslab, 2026*).

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

# Appendix 1

**Appendix 1—table 1.** Response properties of C2 and C3.

Statistical summary of Wilcoxon rank sum test. Sample Size (N) is given in number of cells (C).

| Figure 3 | | Group 1: C2 (magenta) | | | Group 2: C3 (green) | | |
|---|---|---|---|---|---|---|---|
| | p-value G1-G2 | Mean | Std | N | Mean | Std | N |
| Timing ON peak Elevation (e) | 0.0139 | –0.056 | 0.02 | 18 | –0.08 | 0.02 | 10 |
| Timing ON peak Azimuth (e) | 0.0033 | –0.054 | 0.03 | 13 | –0.08 | 0.03 | 25 |
| FWHM Elevation (e) | 0.1300 | 19.697 | 8.82 | 18 | 15.52 | 5.95 | 10 |
| FWHM Azimuth (e) | 0.1192 | 17.769 | 4.07 | 13 | 15.30 | 2.46 | 25 |

**Appendix 1—table 2.** Response properties of Mi1.

Statistical summary of Wilcoxon rank sum test. Sample Size (N) is given in number of cells (C).

| Figure 4 | | Group 1: Control (gray) | | | Group 2: C2 block (magenta) | | |
|---|---|---|---|---|---|---|---|
| | p-value G1-G2 | Mean | Std | N | Mean | Std | N |
| Timing ON peak | 0.0000 | –0.113 | 0.15 | 59 | –0.195 | 0.23 | 73 |
| FWHM | 0.8332 | 16.366 | 5.65 | 59 | 15.447 | 5.01 | 73 |

**Appendix 1—table 3.** Response properties of T4 and T5 to full-field flashes.

Statistical summary of ANOVA.

**T4-ONResps (Figure 5b, Figure 5—figure supplement 1d)**

| Source | SS | dF | MS | F | Prob >F |
|---|---|---|---|---|---|
| Groups | 2.39505 | 3 | 0.79835 | 4.26 | 0.012 |
| Error | 6.18794 | 33 | 0.18751 | | |
| Total | 8.58299 | 36 | | | |

**T5-OFFResps (Figure 5b, Figure 5—figure supplement 1d)**

| Source | SS | dF | MS | F | Prob >F |
|---|---|---|---|---|---|
| Groups | 0.10164 | 3 | 0.03388 | 0.86 | 0.4715 |
| Error | 1.30002 | 33 | 0.03939 | | |
| Total | 1.40166 | 36 | | | |

**T5-ONResps (Figure 5—figure supplement 1e)**

| Source | SS | dF | MS | F | Prob >F |
|---|---|---|---|---|---|
| Groups | 0.57289 | 3 | 0.19096 | 5.02 | 0.0056 |
| Error | 1.25588 | 33 | 0.03806 | | |
| Total | 1.82877 | 36 | | | |

**# Cells T4 (Figure 5—figure supplement 1b)**

| Source | SS | dF | MS | F | Prob >F |
|---|---|---|---|---|---|
| Groups | 4868.62 | 3 | 1622.87 | 13.74 | 4.95355e-06 |
| Error | 4017.2 | 34 | 118.15 | | |
| Total | 8885.82 | 37 | | | |

**# Cells T5 (Figure 5—figure supplement 1b)**

*Appendix 1—table 3 Continued on next page*

*Appendix 1—table 3 Continued*

**T4-ONResps (Figure 5b, Figure 5—figure supplement 1d)**

| Source | SS | dF | MS | F | Prob >F |
|---|---|---|---|---|---|
| Source | SS | dF | MS | F | Prob >F |
| Groups | 163.53 | 3 | 54.5101 | 2.01 | 0.1313 |
| Error | 923.02 | 34 | 27.1477 | | |
| Total | 1086.55 | 37 | | | |

**Appendix 1—table 4.** Flash responses in T4/T5 upon blocking C2 or C3.
Statistical summary of multi comparisons with Bonferroni correction. Sample Size (N) is given in number of flies (F).

| Figure 5 | p-value G1-G2 | p-value G1-G3 | p-value G2-G3 | G1: Control (gray) | | | G2: C2 block (magenta) | | | G3: C3 block (green) | | |
|---|---|---|---|---|---|---|---|---|---|---|---|---|
| | | | | Mean | Std | N (F) | Mean | Std | N (F) | Mean | Std | N (F) |
| Steps dF/F T4 ON (*Figure 5b*) | 0.02745 | 0.02156 | 1 | 0.1908 | 0.11 | 9 | 0.8310 | 0.65 | 8 | 0.8146 | 0.27 | 10 |
| Steps dF/F T5 OFF (*Figure 5b*) | 1 | 1 | 1 | 0.3167 | 0.13 | 9 | 0.3999 | 0.22 | 8 | 0.4350 | 0.18 | 10 |
| Time to peak (*Figure 5c*) | - | - | 0.00032 | - | - | - | 0.4250 | 0.07 | 8 | 1.0500 | 0.38 | 10 |
| Decay rate (*Figure 5c*) | - | - | 0.15063 | - | - | - | 0.6938 | 0.27 | 7 | 0.8685 | 0.2 | 10 |

| Figure 5—figure supplement 1 | p-val G1-G2 | p-val G1-G3 | p-val G1-G4 | G1: Control (gray) | | | G2: C2 block (magenta) | | | G2: C3 block (green) | | | G2: C2C3 block (blue) | | |
|---|---|---|---|---|---|---|---|---|---|---|---|---|---|---|---|
| | | | | Mean | Std | N | Mean | Std | N | Mean | Std | N | Mean | Std | N |
| # Cells T4 (b) | 1 | 0.087025 | 0.000018 | 42.666 | 13 | 9F | 41.666 | 13.6 | 9F | 29.800 | 10.4 | 10 F | 14.800 | 4.49 | 10 F |
| # Cells T5 (b) | 1 | 1 | 1 | 11.000 | 4.55 | 9F | 7.5556 | 6.94 | 9F | 13.400 | 5.35 | 10 F | 10.400 | 3.53 | 10 F |
| Steps dF/F T4 ON (d) | See T1 | See T1 | 0.19447 | 0.1908 | 0.11 | 9F | See T1 | - | - | See T1 | - | - | 0.6352 | 0.51 | 10 F |
| Steps dF/F T5 OFF (d) | See T1 | See T1 | 1 | 0.3167 | 0.13 | 9F | See T1 | - | - | See T1 | - | - | 0.3181 | 0.23 | 10 F |
| Steps dF/F T5 ON(e) | 0.00301 | 0.46976 | 0.76674 | 0.1082 | 0.09 | 9F | 0.4739 | 0.31 | 8F | 0.2711 | 0.12 | 10 F | 0.2483 | 0.19 | 10 F |

**Appendix 1—table 5.** Direction tuning T4 and T5.
Statistical summary of ANOVA. Related to *Figure 6*.

**DS (vector length) all layers: multiway analysis ANOVA**

| Source | SS | dF | MS | F | Prob >F |
|---|---|---|---|---|---|
| Conditions | 2.17782 | 3 | 0.72594 | 60.18 | 0 |
| Layer | 0.13956 | 3 | 0.04652 | 3.86 | 0.0104 |
| T4/T5 | 0.02321 | 1 | 0.02321 | 1.92 | 0.167 |
| Error | 2.34025 | 194 | 0.01206 | | |
| Total | 4.67595 | 201 | | | |

**DS (vector length) averaged across layers T4**

| Source | SS | dF | MS | F | Prob >F |
|---|---|---|---|---|---|

*Appendix 1—table 5 Continued on next page*

*Appendix 1—table 5 Continued*

**DS (vector length) all layers: multiway analysis ANOVA**

| Source | SS | dF | MS | F | Prob >F |
|---|---|---|---|---|---|
| Groups | 0.43445 | 3 | 0.14482 | 29.78 | 2.93772e-08 |
| Error | 0.11671 | 24 | 0.00486 | | |
| Total | 0.55116 | 27 | | | |

**DS (vector length) averaged across layers T5**

| Source | SS | dF | MS | F | Prob >F |
|---|---|---|---|---|---|
| Groups | 827.81 | 3 | 275.937 | 13.14 | 0.0043 |
| Error | 810.19 | 23 | 35.226 | | |
| Total | 1638 | 26 | | | |

**Appendix 1—table 6.** Statistical summary of multi comparisons with Bonferroni correction. Sample Size (N) is given in number of cells (C) or flies (F).

| | p-val G1-G2 | p-val G1-G3 | p-val G1-G4 | G1: Control (gray) | | | G2: C2block (magenta) | | | G2: C3 block (green) | | | G2: C2C3 block (blue) | | |
|---|---|---|---|---|---|---|---|---|---|---|---|---|---|---|---|
| | | | | Mean | Std | N | Mean | Std | N | Mean | Std | N | Mean | Std | N |
| DS T4 (*Figure 6b*) | 0.00000 | 0.00028 | 0.00000 | 0.5165 | 0.08 | 7F | 0.2517 | 0.08 | 8F | 0.3245 | 0.05 | 6F | 0.1852 | 0.03 | 7F |
| DS T5 (*Figure 6b*) | 0.0244 | 0.1894 | 0.0001 | 0.4880 | 0.08 | 7F | 0.3463 | 0.08 | 7F | 0.3822 | 0.10 | 6F | 0.2495 | 0.06 | 7F |
| dF/F −180AD T4 (*Figure 6c*) | 0.0140 | 0.0221 | 0.0262 | 0.6751 | 0.44 | 7F | 1.4116 | 0.70 | 8F | 1.4518 | 0.43 | 6F | 1.1939 | 0.22 | 7F |
| dF/F −135AD T4 (*Figure 6c*) | 0.0205 | 0.0734 | 0.0111 | 0.6233 | 0.40 | 7F | 1.2933 | 0.65 | 8F | 1.2765 | 0.45 | 6F | 1.1196 | 0.30 | 7F |
| dF/F −90AD T4 (*Figure 6c*) | 0.0289 | 0.0734 | 0.0530 | 0.7187 | 0.42 | 7F | 1.3122 | 0.55 | 8F | 1.4167 | 0.54 | 6F | 1.2521 | 0.45 | 7F |
| dF/F −45AD T4 (*Figure 6c*) | 0.9551 | 0.2343 | 0.6200 | 2.2096 | 1.10 | 7F | 2.1420 | 0.91 | 8F | 3.1046 | 1.32 | 6F | 1.7718 | 0.53 | 7F |
| dF/F 0AD T4 (*Figure 6c*) | 0.6943 | 0.1807 | 0.4557 | 3.8594 | 1.93 | 7F | 4.0212 | 1.86 | 8F | 5.3724 | 1.66 | 6F | 2.7353 | 0.82 | 7F |
| dF/F 45AD T4 (*Figure 6c*) | 0.8665 | 0.1375 | 0.7104 | 2.1960 | 1.22 | 7F | 2.2547 | 1.11 | 8F | 3.3790 | 1.15 | 6F | 1.7469 | 0.44 | 7F |
| dF/F 90AD T4 (*Figure 6c*) | 0.0721 | 0.1014 | 0.026 | 0.7076 | 0.48 | 7F | 1.3867 | 0.75 | 8F | 1.6395 | 0.79 | 6F | 1.3073 | 0.36 | 7F |
| dF/F 135AD T4 (*Figure 6c*) | 0.0140 | 0.0513 | 0.0379 | 0.6250 | 0.42 | 7F | 1.3499 | 0.71 | 8F | 1.4048 | 0.56 | 6F | 1.1209 | 0.31 | 7F |
| dF/F −180AD T5 (*Figure 6c*) | 0.0140 | 0.1807 | 0.0023 | 0.6593 | 0.45 | 7F | 1.2500 | 0.34 | 8F | 1.0113 | 0.35 | 6F | 1.5759 | 0.38 | 7F |
| dF/F −135AD T5 (*Figure 6c*) | 0.0401 | 0.2343 | 0.0041 | 0.6537 | 0.43 | 7F | 1.1803 | 0.34 | 8F | 1.0035 | 0.36 | 6F | 1.3635 | 0.35 | 7F |
| dF/F −90AD T5 (*Figure 6c*) | 0.2319 | 0.3660 | 0.0111 | 0.8763 | 0.55 | 7F | 1.3589 | 0.48 | 8F | 1.1756 | 0.30 | 6F | 1.6191 | 0.35 | 7F |
| dF/F −45AD T5 (*Figure 6c*) | 0.4634 | 0.6282 | 0.4557 | 2.0625 | 1.19 | 7F | 2.4539 | 0.96 | 8F | 2.5859 | 0.88 | 6F | 2.7355 | 0.94 | 7F |
| dF/F 0AD T5 (*Figure 6c*) | 0.2810 | 0.6282 | 1 | 3.2665 | 1.45 | 7F | 4.2833 | 1.37 | 8F | 4.1690 | 1.27 | 6F | 3.7727 | 0.97 | 7F |
| dF/F 45AD T5 (*Figure 6c*) | 0.2319 | 0.5338 | 0.8048 | 1.9791 | 1.00 | 7F | 2.7670 | 1.02 | 8F | 2.6985 | 1.32 | 6F | 2.2519 | 0.45 | 7F |
| dF/F 90AD T5 (*Figure 6c*) | 0.2810 | 0.6282 | 0.2086 | 0.8989 | 0.68 | 7F | 1.4130 | 0.54 | 8F | 1.2793 | 0.78 | 6F | 1.4200 | 0.30 | 7F |
| dF/F 135AD T5 (*Figure 6c*) | 0.0939 | 0.1375 | 0.0041 | 0.6688 | 0.45 | 7F | 1.1418 | 0.29 | 8F | 0.9882 | 0.31 | 6F | 1.4702 | 0.41 | 7F |
| *Figure 6—figure supplement 1* | | | | G1: Control (gray) | | | G2: C2block (magenta) | | | G2: C3 block (green) | | | G2: C2C3 block (blue) | | |

*Appendix 1—table 6 Continued on next page*

*Appendix 1—table 6 Continued*

| | p-val G1-G2 | p-val G1-G3 | p-val G1-G4 | G1: Control (gray) | | | G2: C2block (magenta) | | | G2: C3 block (green) | | | G2: C2C3 block (blue) | | |
|---|---|---|---|---|---|---|---|---|---|---|---|---|---|---|---|
| | | | | Mean | Std | N | Mean | Std | N | Mean | Std | N | Mean | Std | N |
| DS Layer 1 T4 | 0.2761 | 1 | 0.0112 | 0.4743 | 0.18 | 7F | 0.2762 | 0.10 | 8F | 0.3346 | 0.08 | 6F | 0.2135 | 0.05 | 6F |
| DS Layer 1 T5 | 0.5212 | 1 | 0.0058 | 0.4786 | 0.12 | 7F | 0.3510 | 0.07 | 7F | 0.3923 | 0.11 | 6F | 0.2220 | 0.09 | 7F |
| DS Layer 2 T4 | 0.0838 | 1 | 0.0147 | 0.4587 | 0.14 | 7F | 0.2411 | 0.13 | 8F | 0.3886 | 0.12 | 6F | 0.1869 | 0.06 | 6F |
| DS Layer 2 T5 | 0.0346 | 0.2933 | 0.0058 | 0.4909 | 0.12 | 7F | 0.3230 | 0.07 | 7F | 0.3695 | 0.11 | 6F | 0.2873 | 0.09 | 7 F |
| DS Layer 3 T4 | 0.0550 | 1 | 0.0064 | 0.5411 | 0.09 | 7F | 0.3467 | 0.11 | 6F | 0.4763 | 0.10 | 5F | 0.2427 | 0.10 | 5F |
| DS Layer 3 T5 | 0.0825 | 0.9529 | 0.0159 | 0.5393 | 0.09 | 5F | 0.3598 | 0.09 | 7F | 0.4398 | 0.12 | 6F | 0.3188 | 0.08 | 6F |
| DS Layer 4 T4 | 0.0631 | 1 | 0.0002 | 0.5944 | 0.12 | 7F | 0.3051 | 0.11 | 6F | 0.4207 | 0.07 | 5F | 0.1721 | 0.10 | 7F |
| DS Layer 4 T5 | 0.3688 | 0.3864 | 0.0178 | 0.5127 | 0.13 | 7F | 0.3291 | 0.17 | 6F | 0.3286 | 0.14 | 5F | 0.2339 | 0.09 | 6F |

**Appendix 1—table 7.** Statistical summary of two-tailed Students's t-tests for *Figure 7* and *Figure 7—figure supplement 1*.

| **Figure 7** | | | G1: *UAS-shi^ts^/+* (blue) | | | G2: *C2-Gal4/+* (cyan) | | | G3: *C2>>Gal4* (Magenta) | | |
|---|---|---|---|---|---|---|---|---|---|---|---|
| | p-value G1-G3 | p-value G2-G3 | Mean | Std | N | Mean | Std | N | Mean | Std | N |
| Turning deceleration (*Figure 7c*) | 0.0018 | 0.0337 | −2.12 | 0.77 | 15 | −1.97 | 2.22 | 18 | −0.20 | 1.96 | 12 |
| % Recovery (*Figure 7f*) | 0.0060 | 0.0033 | 122.55 | 31.55 | 10 | 118.15 | 21.26 | 10 | 82.56 | 25.56 | 10 |
| % Recovery (*Figure 7i*) | 0.0079 | $5.08* \times 10^{-6}$ | 73.60 | 37.12 | 10 | 98.95 | 26.31 | 10 | 34.73 | 17.79 | 10 |
| % Recovery (*Figure 7j*, left panel) | 0.0010 | 0.0269 | 94.57 | 17.88 | 8 | 88.33 | 27.82 | 9 | 61.60 | 13.91 | 8 |
| % Recovery (*Figure 7j*, middle panel) | 0.5644 | 0.8453 | 91.67 | 28.94 | 7 | 80.25 | 17.50 | 9 | 82.63 | 31.40 | 9 |
| % Recovery (*Figure 7k*, left panel) | 0.0269 | 0.2365 | 55.81 | 15.80 | 7 | 42.28 | 19.20 | 9 | 29.59 | 24.25 | 9 |
| % Recovery (*Figure 7k*, middle panel) | 0.9683 | 0.2572 | 48.39 | 13.53 | 7 | 67.05 | 37.16 | 9 | 47.86 | 31.95 | 9 |

| *Figure 7—figure supplement 1* | | | G1: *UAS-shi^ts^/+* (blue) | | | G2: *C3-Gal4/+* (cyan) | | | G3: *C3>>Gal4* (Magenta) | | |
|---|---|---|---|---|---|---|---|---|---|---|---|
| | p-value G1-G3 | p-value G2-G3 | Mean | Std | N | Mean | Std | N | Mean | Std | N |
| Turning deceleration (*Figure 7—figure supplement 1*) | 0.0112 | 0.1016 | −2.12 | 0.77 | 15 | −1.93 | 1.65 | 13 | −0.93 | 1.52 | 16 |

