## [Editor Report · eLife Assessment]

This **important** article reports on the role of specific interneurons in the motion processing circuitry of the fruit fly, and marshals **convincing** evidence from neural recording, genetic manipulation, and behavioral analysis. A significant result ties the activity of C2/C3 neurons to the temporal resolution of the motion vision system. It remains unclear whether disrupting this pathway affects the dynamics of vision more generally.

---

## [Referee Report · Reviewer #1 (Public review)]

Summary:

In this manuscript, Henning et al. examine the impact of GABAergic feedback inhibition on the motion-sensitive pathway of flies. Based on a previous behavioral screen, the authors determined that C2 and C3, two GABAergic inhibitory feedback neurons in the optic lobes of the fly, are required for the optomotor response. Through a series of calcium imaging and disruption experiments, connectomics analysis, and follow-up behavioral assays, the authors concluded that C2 and C3 play a role in temporally sharpening visual motion responses. While this study employs a comprehensive array of experimental approaches, I have some reservations about the interpretation of the results in their current form. I strongly encourage the authors to provide additional data to solidify their conclusions. This is particularly relevant in determining whether this is a general phenomenon affecting vision or a specific effect on motion vision. Knowing this is also important for any speculation on the mechanisms of the observed temporal deficiencies.

Strengths:

This study uses a variety of experiments to provide a functional, anatomical, and behavioral description of the role of GABAergic inhibition in the visual system. This comprehensive data is relevant for anyone interested in understanding the intricacies of visual processing in the fly.

Weaknesses:

The most fundamental criticism of this study is that the authors present a skewed view of the motion vision pathway in their results. While this issue is discussed, it is important to demonstrate that there are no temporal deficiencies in the lamina, which could be the case since C2 and C3, as noted in the connectomics analysis, project strongly to laminar interneurons. If the input dynamics are indeed disrupted, then the disruption seen in the motion vision pathway would reflect disruptions in temporal processing in general and suggest that these deficiencies are inherited downstream. A simple experiment could test this. Block C2, C3, and both together using Kir2.1 and shibiere independently, then record the ERG. Alternatively, one could image any other downstream neuron from the lamina that does not receive C2 or C3 input.

Figure 6c. More analysis is required here, since the authors claim to have found a loss in inhibition (ND). However, the difference in excitation appears similar, at least in absolute magnitude (see panel 6c), for PD direction for T4 C2 and C3 block. Also I predict that C2&C3 block statistically different from C3 only, why? In any case, it would be good to discuss the clear trend in the PD direction by showing the distribution of responses as violin plots to better understand the data. It would be also good to have some raw traces to be able to see the differences more clearly, not only polar plots and averages.

The behavioral experiments are done with a different disruptor than the physiological ones. One blocks chemical synapses, the other shunts the cells. While one would expect similar results in both, this is not a given. It would be great if the authors could test the behavioral experiments with kir2.1 too.

Comments on revisions:

I have no further comments.

---

## [Referee Report · Reviewer #2 (Public review)]

The work by Henning et al. explores the role of feedback inhibition in motion vision circuits, providing the first identification of inhibitory inheritance in motion-selective T4 and T5 cells of Drosophila. Among the strengths of this work is the verification of the GABAergic nature of C2 and C3 with genetic and immunohistochemical approaches. In addition, double-silencing C2&C3 experiments help to establish a functional role for these cells. The authors holistically use the Drosophila toolbox to identify neural morphologies, synaptic locations, network connectivity, neuronal functions and the behavioral output.

A limitation of the study is that the mediating neural correlates from C2&C3 to T4&T5 are not clarified, rather Mi1 is found to be one of them. In the future, the same set of silencing experiments performed for C2-Mi1 could be extended to C2 &C3-Tm1 or Tm4 to find the T5 neural mediators of this feedback inhibition loop. Future experiments might also disentangle the parallel or separate function of C2 and C3 neurons.

In summary, this work advances our current knowledge in Drosophila motion vision and sets the way for further exploring the intricate details of direction selective computations.

Comments on revisions:

A label for T5 is missing from Figure 5b. Thank you for addressing our concerns and considering each of our suggestions.

---

## [Author Response]

The following is the authors’ response to the original reviews.

**Public Reviews:**

**Reviewer #1 (Public review):**
Summary:In this manuscript, Henning et al. examine the impact of GABAergic feedback inhibition on the motion-sensitive pathway of flies. Based on a previous behavioral screen, the authors determined that C2 and C3, two GABAergic inhibitory feedback neurons in the optic lobes of the fly, are required for the optomotor response. Through a series of calcium imaging and disruption experiments, connectomics analysis, and follow-up behavioral assays, the authors concluded that C2 and C3 play a role in temporally sharpening visual motion responses. While this study employs a comprehensive array of experimental approaches, I have some reservations about the interpretation of the results in their current form. I strongly encourage the authors to provide additional data to solidify their conclusions. This is particularly relevant in determining whether this is a general phenomenon affecting vision or a specific effect on motion vision. Knowing this is also important for any speculation on the mechanisms of the observed temporal deficiencies.Strengths:This study uses a variety of experiments to provide a functional, anatomical, and behavioral description of the role of GABAergic inhibition in the visual system. This comprehensive data is relevant for anyone interested in understanding the intricacies of visual processing in the fly.Weaknesses:(1) The most fundamental criticism of this study is that the authors present a skewed view of the motion vision pathway in their results. While this issue is discussed, it is important to demonstrate that there are no temporal deficiencies in the lamina, which could be the case since C2 and C3, as noted in the connectomics analysis, project strongly to laminar interneurons. If the input dynamics are indeed disrupted, then the disruption seen in the motion vision pathway would reflect disruptions in temporal processing in general and suggest that these deficiencies are inherited downstream. A simple experiment could test this. Block C2, C3, and both together using Kir2.1 and Shibire independently, then record the ERG. Alternatively, one could image any other downstream neuron from the lamina that does not receive C2 or C3 input.

Given the prominent connectivity of C2 and C3 to lamina neurons, we actually expected that lamina processing is also affected. We did the experiment of silencing C2 and recording in the lamina neuron L2 and found no significant difference in their response profile (Author response image 1).

**Author response image 1. sa3fig1:** Calcium responses of L2 axon terminals to full field ON and PFF flashes for controls (grey, N=8 flies, 59 cells) or while genetically silencing C2 using *shibirets* (magenta, N=4 flies, 26 cells). Traces show mean +- SEM.

We could include these data in the main manuscript, but we do not really feel comfortable in claiming that C2 and C3 have a specific role in motion processing only, even if it was predominantly affecting medulla neurons. To our knowledge, how peripheral visual circuitry contributes to any other visual behaviors, such as object detection, including the pursuit of mating partners, or escape behaviors, is not well understood. Instead, we added a sentence to the discussion stating that our work does not exclude that, given their wide connectivity, C2 and C3 are also involved in other visual computations.

(2) Figure 6c. More analysis is required here, since the authors claim to have found a loss in inhibition (ND). However, the difference in excitation appears similar, at least in absolute magnitude (see panel 6c), for PD direction for the T4 C2 and C3 blocks. Also, I predict that C2 & C3 block statistically different from C3 only, why? In any case, it would be good to discuss the clear trend in the PD direction by showing the distribution of responses as violin plots to better understand the data. It would also be good to have some raw traces to be able to see the differences more clearly, not only polar plots and averages.

We apologize: The plots in the manuscript show the mean across all cells, but the statistics were done more conservatively, across flies. We corrected this mismatch and the figure now shows the mean ± ste across flies after first averaging across cells within each fly. Thank you for pointing this out. Since we recorded n=6-8 flies per genotype, we did not include violin plots, which would indeed make sense if we showed data for each cell.

(3) The behavioral experiments are done with a different disruptor than the physiological ones. One blocks chemical synapses, the other shunts the cells. While one would expect similar results in both, this is not a given. It would be great if the authors could test the behavioral experiments with Kir2.1, too.

We have tried this experiment, but unfortunately, flies were not walking well on the ball, and we were not able to obtain data of sufficient quality.

**Reviewer #2 (Public review):**
Summary:The work by Henning et al. explores the role of feedback inhibition in motion vision circuits, providing the first identification of inhibitory inheritance in motion-selective T4 and T5 cells of Drosophila. This work advances our current knowledge in Drosophila motion vision and sets the way for further exploring the intricate details of direction-selective computations.Strengths:Among the strengths of this work is the verification of the GABAergic nature of C2 and C3 with genetic and immunohistochemical approaches. In addition, double-silencing C2&C3 experiments help to establish a functional role for these cells. The authors holistically use the Drosophila toolbox to identify neural morphologies, synaptic locations, network connectivity, neuronal functions, and the behavioral output.Weaknesses:The authors claim that C2 and C3 neurons are required for direction selectivity, as per the publication's title; however, even with their double silencing, the directional T4 & T5 responses are not completely abolished. Therefore, the contribution of this inherited feedback in direction-selective computations is not a prerequisite for its emergence, and the title could be re-adjusted.

We adjusted the title to “are involved in motion detection.”

Connectivity is assessed in one out of the two available connectome datasets; therefore, it would make the study stronger if the same connectivity patterns were identified in both datasets.

We did not assume large differences between the datasets because Nern et al. 2025 described no major sexual dimorphism. To verify this, we now plotted C2 and C3 connectivity from the three major EM datasets that include C2/C3 connectivity, the female FAFB dataset (Zheng et al. 2018, Dorkenwald et al. 2024, Schlegel et al. 2024) the male visual system (Nern et al. 2025), and the 7-column dataset (Takemura et al. 2015) and see no major differences (Author response image 2 and Author response image 3).

**Author response image 2. sa3fig2:** Relative pres- and post-synaptic counts for C3 from 3 different data sets. Shown are up to ten post- or pre-synaptic partner neurons.

**Author response image 3. sa3fig3:** Relative pres- and post-synaptic counts for C2 from 3 different data sets. Shown are up to ten post- or pre-synaptic partner neurons.

The mediating neural correlates from C2 & C3 to T4 & T5 are not clarified; rather, Mi1 is found to be one of them. The study could be improved if the same set of silencing experiments performed for C2-Mi1 were extended to C2 &C3-Tm1 or Tm4 to find the T5 neural mediators of this feedback inhibition loop. Stating more clearly from the connectomic analysis, the potential T5 mediators would be equally beneficial. Future experiments might also disentangle the parallel or separate functions of C2 and C3 neurons.

We fully agree that one could go down this route. Given the widespread connectivity of C2 and C3, and the fact that these are time-consuming experiments with often complex genetics, we had decided to instead study the “compound effect” of C2 and C3 silencing by analyzing T4/T5 physiological properties and motion-guided behavior. We now explicitly explain this logic by saying, “To understand the compound effect of C2 and C3 on motion processing, we focused on the direction-selective T4/T5 neurons, which are downstream of many of the neurons that C2 and C3 directly connect to.”

Finally, the authors' conclusions derive from the set of experiments they performed in a logical manner. Nonetheless, the Discussion could benefited from a more extensive explanation on the following matters: why do the ON-selective C2 and C3 neurons control OFF-generated behaviors, why the T4&T5 responses after C2&C3 silencing differ between stationary and moving stimuli and finally why C2 and not C3 had an effect in T5 DS responses, as the connectivity suggests C3 outputting to two out of the four major T5 cholinergic inputs.

Apart from the behavioral screen results, we only tested ON edges in our more detailed behavioral characterizations. And while we show phenotypes for the OFF-DS cell T5, it is well established that inhibitory cells that respond to one contrast polarity can function in the pathway with the opposite contrast polarity (e.g., the OFF-selective Mi9 in the ON pathway). We realized that our narrative in the results section was misleading in this regard (we had given the ON selectivity of C2/C3 as one argument why we first focused on the ON pathway) and eliminated this argument.

For the differential involvement of C2/C3 for T4/T5 responses to stationary and moving stimuli (C2 and C3 silencing affects both T4 and T5 DS responses, but mostly T4 flash responses): We mostly took the disinhibition of flash responses in T4 as a motivation to look more specifically at a potential role in motion-computation. We now added a sentence about the potential emergence of these flash responses to the already extensive discussion paragraph “How could inhibitory feedback neurons affect motion detection in the ON pathway?”

Last, we added a discussion point about the relationship between C2 and C3 connectivity and the functional consequences, and discussed the fact that C3 connectivity alone does not correlate with a functional role of C3 (alone) in DS computation.

**Reviewer #3 (Public review):**
Summary:This article is about the neural circuitry underlying motion vision in the fruit fly. Specifically, it regards the roles of two identified neurons, called C2 and C3, that form columnar connections between neurons in the lamina and medulla, including neurons that are presynaptic to the elementary motion detectors T4 and T5. The approach takes advantage of specific fly lines in which one can disable the synaptic outputs of either or both of the C2/3 cell types. This is combined with optical recording from various neurons in the circuit, and with behavioral measurements of the turning reaction to moving stimuli.The experiments are planned logically. The effects of silencing the C2/C3 neurons are substantial in size. The dominant effect is to make the responses of downstream neurons more sustained, consistent with a circuit role in feedback or feedforward inhibition. Silencing C2/C3 also makes the motion-sensitive neurons T4/T5 less direction-selective. However, the turning response of the fly is affected only in subtle ways. Detection of motion appears unaffected. But the response fails to discriminate between two motion pulses that happen in close succession. One can conclude that C2/C3 are involved in the motion vision circuit, by sharpening responses in time, though they are not essential for its basic function of motion detection.Strengths:The combination of cutting-edge methods available in fruit fly neuroscience. Well-planned experiments carried out to a high standard. Convincing effects documenting the role of these neurons in neural processing and behavior.Weaknesses:The report could benefit from a mechanistic argument linking the effects at the level of single neurons, the resulting neural computations in elementary motion detectors, and the altered behavioral response to visual motion.

We agree that we cannot fully draw this mechanistic argument, but we also do not think that this is a realistic goal of this study. Even in a scenario where one would measure the temporal and spatial properties of “all” neurons that are connected to C2 and C3, this would likely not reveal the full mechanisms linking the single neurons to DS computation, but would require silencing specific connections, or specific molecular components of the connection, or could be complemented by models. A beautiful example where such a mechanistic understanding was achieved, recently published in Nature, essentially focused on a single synaptic connection (between Mi9 and T4) (Groschner et al. 2024), and built on extensive work that had already highlighted the importance of these neurons. We would further argue that the field does not have a good understanding of how T4/T5 responses are translated into behavior. Although possible pathways emerge from connectomes, it is for example not understood why the temporal frequency tuning of T4/T5 substantially differs from the temporal frequency tuning of the optomotor response.

We therefore would like to highlight that the focus of our study was not to connect all those pieces, but rather to highlight the hitherto unknown overall importance of inhibitory feedback neurons for visual computations along the visual hierarchy, from individual neuron properties, via DS computation, to the temporal precision of the optomotor response.

**Recommendations for the authors:**

**Reviewer #1 (Recommendations for the authors):**
(1) Line 52: "The functional significance of feedback neurons, particularly inhibitory feedback mechanisms, in early visual processing is not understood."This is incorrect not only because it is referred to as a general statement, but also because many studies have examined inhibition in flies. It may not be solely GABAergic inhibition, but that is just one type. While some discussions later address feedback from horizontal cells in the retina, etc., there is no mention of work on color vision, which requires feedback. Please rephrase.

We now say “visual motion processing” in this sentence, and added a sentence on color vision: “... color-opponent signalling requires reciprocal inhibition between photoreceptors as well as feedback inhibition from distal medulla (Dm) neurons. (Schnaitmann et al., 2018, Heath et al., 2020, Schnaitmann et al., 2024). “

(2) Line 197: "Because a previous studies" One or many?, but more important, please cite them.

We corrected to “a previous study” and cite Tuthill et al. 2013

(3) Line 172: I noticed a few minor grammatical errors and wording issues, such as the use of "we next" twice in one sentence. "To next identify potential GABAergic neurons that are important for motion computation in the ON pathway, we next intersected 12 InSITE-Gal4." I am bad at picking them out, but since I noticed them, I would strongly suggest looking at the text carefully again.

We deleted one occurrence of ‘next’, thank you for catching that.

(4) Question to the authors. Why did you use twice independent lines and not checkers for the white noise analysis in Figure 3e?

We used flickering bars because many visual system neurons tested in our lab respond with a better signal-to-noise ratio as compared to checkerboards. Flickering bars also appear to be more suited to isolate the spatial surround of neurons. This type of stimulus has been successfully used in previous studies to extract receptive fields of neurons in the fly visual system (Arenz et al. 2017; Leong et al., 2016, Salazar-Gatzimas et al. 2016; Fisher et al. 2015, …).

(5) Line 248: "Because C2 emerged as a prominent candidate from the behavioral screen, we focused on C2 and asked how silencing C2 affects..." Please state how here. I would need to go to the methods.

We added a sentence “C2 was silenced by expression of *UAS-shibirets* (*UAS-shits*) for temporal control of the inhibition of synaptic activity.”

(6) Much of the work in the blowfly uses picrotoxinin to block GABAergic inhibition in the visual motion pathway. It would be useful to mention some of this early work and its results, particularly that of Single et al. (1997). It might be interesting to reinterpret their results.

Thank you for pointing this out. We added this paragraph to the discussion: ‘Work in blowflies has found a severe impact of GABAergic signaling for DS in LPTCs downstream of T4 and T5 cells, using application of picrotoxin to the whole brain (Single et al. 1997; Schmid and Bülthoff 1988). Although the loss of DS in LPTCs could originate from direct inhibitory synapses onto LPTCs (Mauss et al. 2015; Ammer et al. 2023), the disruption of GABAergic signaling in upstream circuitry, which reduces DS in T4 and T5, may also contribute to the phenotype seen in LPTCs.’

**Reviewer #2 (Recommendations for the authors):**
The following set of corrections aims to better the scientific and presentation aspects of this work.(1) The title of the work implies that C2 and C3 neurons are required for motion processing, whereas the study shows their participation in motion computations, which persists post their silencing. Therefore, "Inhibitory columnar feedback neurons contribute to Drosophila motion processing" would be a more appropriate title.

We rephrased the title to say that inhibitory feedback neurons “are involved in” motion processing.

(2) The morphology of C2 and C3 neurons, i.e., ramifications in medulla & cell body in medulla and axonal targeting to lamina, implies their feedback role. It would be important to mention the specific feedback loop they participate in and the role of Mi1 more extensively in lines 36, 120.

We find it hard to speculate on the specific feedback loops that C2 and C3 are involved in from their widespread input and output connectivity. If we had, we would have wanted to support this by functional measurements of this specific loop, which was not the goal of this study.

(3) In lines 55-89, the authors explore the instances of feedback inhibition within and across species and modalities. For the Drosophila visual example (lines 76-89), given that it also addresses motion circuits, the following studies should be included:Ammer, G., Serbe-Kamp, E., Mauss, A.S., et al. Multilevel visual motion opponency in Drosophila. Nat Neurosci 26, 1894-1905 (2023). https://doi.org/10.1038/s41593-023-01443-z. Mabuchi Y, Cui X, Xie L, Kim H, Jiang T, Yapici N. Visual feedback neurons fine-tune Drosophila male courtship via GABA-mediated inhibition. Curr Biol. 2023 Sep 25;33(18):3896-3910.e7. doi: 10.1016/j.cub.2023.08.034.

We added a sentence on the Ammer et al. finding to the introduction. Since the introduction paragraph focuses on known physiological effects within the visual system, we did not find a good fit for the Mabuchi et al. study, which focuses on serotonergic feedback neurons with a role far downstream in courtship behavior.

(4) In lines 102-103, the following work should be referenced: Groschner LN, Malis JG, Zuidinga B, Borst A. A biophysical account of multiplication by a single neuron. Nature. 2022 Mar;603(7899):119-123. doi: 10.1038/s41586-022-04428-3.

We cited a few of the many papers that used “modeling frameworks” and selected the ones focusing on the entire feedforward circuitry. To also give credit to the Borst lab, we instead added Serbe et al. 2016 here.

(5) In lines 107-108, the Braun et al. (2023) study has not performed Rdl knockdown experiments in T4 cells; hence, it needs to be better clarified in the text.

We corrected this in the text.

(6) Even though the dataset was previously published, a summary plot of the different phenotypes would be very helpful to the reader. Moreover, in line 131, as the study focuses on motion vision, it would be better to use "early motion visual processing" rather than "early visual processing.”

We added a summary plot of the behavioral screen data to Supplementary figure 1, and rephrased previous line 131.

(7) The first result section title excludes C3 neurons, even though in lines 172-179 they are addressed; therefore, the C3 inclusion is suggested as in "GABAergic C2 and C3 neurons control behavioral responses to motion cues". The term "required" should be excluded from the title as the other neuronal types encountered in the InSITE drivers were never quantified; thus, the "behavioral requirement" might come from these other neurons as well.

From the experiments shown in this paragraph alone we cannot make conclusive claims about C3, as it was also weakly visible in one of our genetic control in the intersectional strategy that we took (we had written: “This strategy also revealed other GABAergic cell types, including the columnar neuron C3 and the large amacrine cell CT1 which were however also weakly present in the *gad1-p65AD* control).

We changed the title of this paragraph to: A forward genetic behavioral screen identifies GABAergic C2 neurons to be involved in motion detection.

(8) In line 142, it should be clearly stated that the MultiColor FlpOut technique was used and should also be cited: Nern A, Pfeiffer BD, Rubin GM. Optimized tools for multicolor stochastic labeling reveal diverse stereotyped cell arrangements in the fly visual system. Proc Natl Acad Sci U S A. 2015 Jun 2;112(22):E2967-76. doi: 10.1073/pnas.1506763112.

We did not use MCFO clones, but simple Flp-out clones, and the genotype and reference for this were given in the methods: UAS-FRT-CD2y+-RFT-mCD8::GFP; UAS-Flp , (Wong et al. 2002). To make this clearer, we now also cite (Wong et al. 2002) in the results section.

(9) In Figure 1c, a description of RFP should be written as it is already in Supplementary Figure 1c.

We added this to the Figure caption.

(10) In line 172, "next" is redundant as it was previously used at the beginning of the sentence.

Removed

(11) In line 175, based on both figures that the authors refer to, instead of C2, C3 should be written.

We do indeed see C3 labeled in the images, but also in a gad1-p65AD control. We thus cannot be sure if C3 indeed reflects the intersection pattern. However, the three lines shown in Figure 1d clearly also label C2, which is not seen in the control condition.

(12) In line 184, a split-C2 line is used (and a split C3 as in Supplementary Figure 2). It would enhance the credibility of the work and even be appropriate afterwards to use the word "requirement" if this split-C2 line was used for behavioral experiments, as in Gohl et al., 2011, and Sillies et al.,2013 studies.

We are indeed using the same split-C2 line for imaging and for behavioral experiments in Figure 7. We see Figure 1 (and with that, Silies et al. 2013) as a first pass screen, from which we obtained candidates, which we then more thoroughly tested throughout the remaining manuscript, with more specific lines. We are no longer using the word “requirement”

(13) In lines 186-188, is DenMark used as a postsynaptic marker? If yes, an additional control would be the use of Discs-large (DLG) as a postsynaptic marker, as DenMark would not be restricted to postsynaptic densities.

Yes, we used DenMark as written in the sentence “we expressed GFP-tagged Synaptotagmin (Syt::GFP) to label pre-synapses together with the dendritic marker DenMark (Nicolai et al., 2010)”. Since our claims about widespread C2 and C3 connectivity are further supported by connectomics, we did not use another postsynaptic marker.

(14) In line 191, L2 is mentioned as presynaptic, whereas in Figure 2b is clearly postsynaptic.

We write “This revealed that C2 forms several presynaptic contacts with the lamina neurons L5, L1, and L2” . L5, L1, and L2 are hence postsynaptic to C2, which is what is plotted in Figure 2b.

(15) In line 197, the "a" in "because a previous studies" should be removed, and these studies should be cited as the authors do in line 514.

Done as suggested.

(16) In line 1191, the figure title uses the term "required", whereas the plotted data suggest that T4 and T5 responses remain DS after C2&C3 silencing. Rephrasing to "C2 and C3 affect direction-selective.." would be better suited.

We replaced “required” with “contribute to”

(17) In the legend of Figure 2b, the "Counts of synapses" is misleading. The number plotted refers to the percentage of synapse counts from the target neuron.

Corrected.

(18) A general question about the C2 and C3 ON selectivity: How would the authors explain the OFF deficits from the published behavioral screening in Supplementary Figure 1a? Do the other InSITE neurons contribute to it? This needs to be further elaborated in the discussion.

A neuron being ON selective does not imply that it is functionally required in the ON pathway only. In fact, Mi9, a major component of the ON pathway (even if not “required” under many stimulus conditions), is OFF selective.

Furthermore, both we (Ramos-Traslosheros and Silies, 2021) and others (Salazar-Gatzimas et al. 2019) have shown that both ON and OFF signals are combined in ON and OFF pathways, which is further supported by connectomics data. We clarified the transition from physiology to function in the results section, as already explained above.

(19) In line 216, the authors' image from layer M1, but the reasoning behind this choice is missing. The explanation gap intensifies after you proceed with further examining the layer-specific responses in Supplementary Figure 2. Is this because C2 and C3 receive their inputs in M1, as is insinuated in line 219?

As Supplementary Figure 2 shows, we initially imaged from all layers of the medulla, where C2 arborizes. Because the response properties, including kinetics, weren’t different, we had no reason to believe that C2 is highly compartmentalized. We thus subsequently focused on layer M1, where amplitudes were highest. We clarified this in the text.

(20) In line 229, it should be clear whether the STRFs come from M1 measurements. STRF analysis in M5, M8, and M9/10 also verifies that the C2, C3 multicolumnar span would further strengthen the results. Given the focus of the work in Mi1 and T4/T5, Mi1-C2 connections should be clarified in terms of which medulla layer they formulate. Additionally, the reasoning behind showing in Figure 3 STRFs from M1 measurements, even though Supplementary Figure 2b implies equal responses in M9/10, where also Tm1 and Tm4 output from C3, should be explained.

We never recorded STRFs in the silenced condition and make no claims about C2 changing spatial properties of Mi1. We added the information that STRFs were recorded in layer M1 to the figure caption. We checked the specific connectivity of C2 and Mi1 and they indeed connect in M1 (Author response image 4), but regardless of this result, there is no evidence for compartmentalization in these columnar neurons.

**Author response image 4. sa3fig4:** Image of a C2 (blue) and Mi1 (yellow) neuron from EM Data (FAFB). Circles depict synapses from C2 to Mi1 in layer M1 of the medulla.

(21) In Figure 3e, the statistical significance or lack thereof is not visible at the bar plot.

Consistently throughout the manuscript, we now just indicate if a comparison is significant. If nothing is shown, it means that it is not.

To clarify this, we added a sentence to the statistics section in the methods now saying: We show significant differences in figures using asterisks (p<0.05 *,p<0.01 **, p<0.001***). Non-significant differences are not further indicated.

Please note that based on another reviewer comment, we also adapted the analysis of the kernels. This changed the statistics to be significant for the timing of the on peak response (Figure 3e).’

(22) In line 249, it is mentioned that the strongest C2 connection is Mi1; this does not derive from the data shown in Figure 2b.

We intended to look at medulla neurons, and Mi1 is the most connected medulla neuron to C2. We clarified that in the text, which now reads: “Because C2 emerged as a prominent candidate from the behavioral screen, we focused on C2 and asked how silencing C2 affects temporal and spatial filter properties of the medulla neurons that provide direct input to T4 neurons. We chose to test Mi1 as it is the medulla neuron most strongly connected to C2.”

(23) The result section title "C2 & C3 neurons shape response properties of the ON pathway medulla neuron Mi1" does not include C3 results. This would be fundamental to have. As previously mentioned, the neural correlates of this inhibitory feedback loop should be clearly defined, and the current version of this work evades doing so.

We corrected the title. As discussed elsewhere, it was not the goal of this study to work the specific contributions of C2 (and C3) to all neurons they connect to, but rather focus on the compound effect for motion detection.

(24) In line 276, the following work should be cited: Maisak MS, Haag J, Ammer G, Serbe E, Meier M, Leonhardt A, Schilling T, Bahl A, Rubin GM, Nern A, Dickson BJ, Reiff DF, Hopp E, Borst A. A directional tuning map of Drosophila elementary motion detectors. Nature. 2013 Aug 8;500(7461):212-6. doi: 10.1038/nature12320.

We added the citation.

(25) In line 273, the title implies the investigation of the spatial filtering of T4 and T5 cells. This does not take place in the respective result section.

We changed the title to: “C2 and C3 shape temporal and spatial response properties of T4 and T5 neurons.”

(26) In line 280, Kir2.1 is used, whereas previously thermogenetic silencing with Shibirets was preferred; could the authors elaborate on this choice in the text, for example, genetic reasons?

We generally prefer shibire[ts] because of its inducible nature. However, our T4/T5 recordings too included more stimuli (motion stimuli) than the Mi1 recordings, and the effect of shi[ts] mediated silencing by pre-heating the flies (as established by Joesch et al. 2010) was not longlasting enough for these experiments, which is why we used Kir2.1. In a previous set of experiments, we had tried incubating flies while imaging, but this induced too large movements of the brain and T4/T5 recordings were not stable enough.

(27) In lines 290-291, T5 ON suppression is found to be affected by C2 silencing, but the bar plot in Figure 5b uses the OFF-step data. It would be best if the ON-step data for T5 cells were also plotted.

ON-step data for T5 are plotted in Supplementary Fig. 3e

(28) In line 288, "when C2 was also blocked", "also" should be included, as you are referring to double silencing.

Sorry for the confusion, we called the wrong figure in that sentence. Here, we wanted to point at the increased response of T4 to the ON-step upon C2 silencing, which was quantified in Supplementary Fig. 3e.

(29) In line 312, it is important to mention in the discussion why it is the case that C2 and not C3 had an effect on T5 DS responses. C2 outputs to Tm1, whereas C3 to Tm1 and Tm4, based on Figure 2b, with Tm1 and Tm4 being one of the four major cholinergic T5 inputs. Hence, it would be natural to think that C3 and not C2 would affect T5 responses.

We addressed this in the discussion.

(30) In lines 326-328, it is crucial to mention the neural correlates that connect C2 and C3 to T4 and T5. Additionally, the Shinomiya et al. (2019) study shows C3 to T4 connections, which are mentioned in the discussion and should be cited in line 429.

We do not think that mentioning neural correlates at this point is crucial, as these sentences were concluding a paragraph in which we link C2/C3 silencing to T4/T5 responses. We also do not know the neural correlates (but for Mi1) so this would not be accurate.

We have been mentioning C3 to T4 connection in both the results and discussion, and our analysis (Figure 2) stems from the FAFB dataset. We added citations to both results and discussion.

(31) In Figure 6a, compared to Figure 3b, the term compass plots is used instead of polar plots. It would be best to use one consistent term. Additionally, in Figure 6c, it is not mentioned if the responses across genotypes are the outcome of averaging across subtype responses.

These two plots are not the same; a compass plot is a sub-category of polar plots. Polar plots, as in Figure 3, show the response amplitude of the neurons to the different directions of motion. Instead, compass plots, as in Figure 6, show vectors that depict the tuning direction and the strength of tuning of individual neurons.

We added the following sentence to clarify the calculation in Figure 6c: ‘To average responses of all neurons, the PD of each neuron was determined by its maximal response to one of 8 directions shown.'

(32) In line 344, the title could be adjusted to "C2 is controlling the temporal dynamics of ON behavior", under the same reasoning of 'requirements' explained before.

We think that “is controlling” is a stronger claim than “being required”. For a geneticist, the word “required” simply means that there is a(ny) loss of function phenotype, i.e., a reduction in DS when C2 and C3 are silenced/blocked. Many neurons are sufficient but not required to induce a certain behavior (i.e., they can induce a behavior when ectopically activated, but show no significant loss of function phenotype). We therefore consider it remarkable that C2 and C3 silencing indeed shows a significant reduction in DS.

However, we do not want to overclaim anything, and the title now reads: “T4 tunes the temporal dynamics of ON behavior”

(33) In Figure 7c, the plot legend should be "deceleration".

Corrected

(34) In line 424, the Braun et al. (2023) experiments were performed in T5 cells as previously mentioned.

Corrected

(35) In line 435, the authors mention that both ON-selective C2 and C3 neurons act partially in parallel pathways. In Figure 2b, the upstream circuitry between C2 and C3 is identical. How would they explain the functional-connectivity contradiction?

In terms of acting in parallel pathways, downstream, not upstream, connectivity of C2 and C3 will matter, which is not identical. C2 for example connects to Mi1, L1, and L4, whereas C3 does not. On the other hand, C3 connects to Mi9 and Tm4, which C2 does not.

(36) In lines 445-447, the authors address C2 and C3 neurons as columnar, whereas they previously showed in Figure 3 that they are multicolumnar.

Here, we refer to the nomenclature of Nern et al, that use the term “columnar” whenever something is present in each column. We specifically define this by saying “only 15 cells are truly columnar in the sense that they are present once per column and present in each column”. In the results section, we instead talk about “functionally multicolumnar” and changed a sentence in the discussion to say “The spatial receptive fields of C2 and C3 are consistent with the multicolumnar branching of their projections in the medulla” to avoid any such confusion.

(37) In line 448, "thus" is repetitive, and the extracted view in line 449 does not contribute to the essence of the study.

Fixed.

(38) In line 459, the authors refer to inhibition inheritance; this term should be used frequently in the text in case the neural correlates between C2 & C3 and T4 & T5 are not deciphered.

We think this point is very clear throughout the manuscript now. As one prominent example, we added a sentence to the first paragraph of the discussion saying “Given the widespread connectivity of C2 and C3 to neurons upstream of T4/T5, this effect [on DS tuning] is likely inherited from upstream neurons of T4/T5.”

(39) In line 521, the transition between sentences is problematic.

Corrected

(40) For Supplementary Figure 1, why were the ON-motion deficits not addressed with the antibody approach used for Supplementary Figure 1a?

The approach using anti-GABA stainings turned out to be largely redundant with the intersectional strategy. Furthermore, the intersectional strategy provided the full morphology of the cell and, hence, led to easier identification of the cell types involved.

(41) In line 1169, C2 is mentioned, whereas C3 is annotated in the figure.

Corrected

(42) A general comment is that Tm1 inputs could be a good candidate for assessing T5 inputs, as performed for Mi1-T4 in Fig.4. Such experiments would enhance the understanding of inhibitory inheritance to T5 responses.

We fully agree.

(42) Do the authors have any indication or experiments done regarding the C2&C3 role in T4&T5 velocity tuning? This would be complementary to the direction of this study.

This is a good idea, that we had tried. However, we did not see a difference between control and C2 silencing for the temporal frequency tuning of T4/T5. As velocity is closely related to temporal frequency tuning, we would not expect to see a difference there either.

While it would have been nice to be able to draw such a link, we would also state that our behavioral data are a bit different: We did not look at temporal frequency tuning per se, and overall, it is not well understood how responses in T4/T5 relate to behavior, as they for example have different frequency tunings (T4/T5 physiology: Maisak et al., 2013, Arenz et al., 2017; optomotor behaviour: Strother et al.,2017, Clark et al., 2013).

(43) As a suggestion, Figure 7 would be better positioned as Figure 4, right after the ON-selectivity finding of C2 neurons.

We preferred to keep the current order.

**Reviewer #3 (Recommendations for the authors):**
Main recommendation:It would be useful to propose a neural circuit model that connects the various observations. One can draw here on the many circuit models for motion vision in the prior literature.(1) How might the extended response in upstream neurons Mi1 lead to the inappropriate nulldirection responses in T4/T5?

This is a good question and we can only speculate. Mi1 responses are enhanced upon C2 silencing and T4 responses to full field flash responses are also enhanced. Likely, these motionindependent responses are also seen when the edge travels into the non-preferred direction, whereas this non-motion response would likely be masked by the motion response to the preferred direction. The phenotype seen in T5 is likely inherited from medulla neurons, e.g. Tm1, to which C2 connects. How the delay of the Mi1 response upon C2 silencing may specifically affect ND responses, we don’t know.

(2) How is the loss of DS in T4/T5 compatible with the continued sensitivity to motion in the turning response? Perhaps the signal from 180-degree oppositely tuned T-cells gets subtracted, so as to remove the baseline activity?

This is a great question that we cannot answer. Overall, perturbations that affect T4/T5 physiology do not necessarily manifest in equivalent phenotypes when looking at behavioral turning responses. Prominent examples come from silencing core neurons of motion-detection circuits, such as Mi1 and Tm3 (see Figure 4, Strother et al. 2017).

(3) How do the altered dynamics in upstream neurons relate to the loss of high-frequency discrimination in the behavior? One would want to explain why the normal fly has a pronounced decay in the response even though the motion is still ongoing (Figure 7b left, starting at 0.4 s). That decay is missing in the mutant response.

That is an excellent question that we unfortunately do not have an answer for. Please note that our visual stimuli is a single edge which is sweeping across the eye, and which might not elicit equally strong responses at each position of the eye, or each time during the stimulus presentation.

In terms of linking the dynamics of upstream neurons to behavior, we already pointed out above that it is not well understood how responses in T4/T5 relate to behavior, as they for example have different frequency tuning, with T4/T5 neurons being tuned to lower temporal frequencies than the turning behavior of a fly walking on a ball (T4/T5 physiology: Maisak et al., 2013, Arenz et al., 2017; optomotor behaviour: Strother et al.,2017, Clark et al., 2013).

Other recommendations:(1) Abstract line 37 "At the behavioral level, feedback inhibition temporally sharpens responses to ON stimuli, enhancing the fly's ability to discriminate visual stimuli that occur in quick succession." It may be worth specifying *moving* stimuli.

Done as suggested

(2) Line 52: "The functional significance of feedback neurons, particularly inhibitory feedback mechanisms, in early visual processing is not understood." This seems overly negative. Subsequent text mentions a number of such instances that are understood, and one could add more from the retina.

We agree. We rephrased to say ‘motion vision’ and added more examples of known roles of feedback inhibition

(3) Line 69: "inhibitory feedback signals from horizontal cells and amacrine cells to photoreceptors and bipolar cells, respectively, are involved in multiple mechanisms of retinal processing, including global light adaptation, spatial frequency tuning, or the center-surround organization (Diamond 2017)." Maybe add the proven role in temporal sharpening of responses, which is of relevance to the present report.

We added temporal sharpening to that introduction point.

(4) Figure 1: The text for this figure talks about behavioral motion detection deficits in various lines. Maybe add an example of the behavioral effects to this figure.

We added a summary plot of the behavioral screen data to Supplementary figure 1.

(5) Line 325: "the timing of the ON peak tended to be slower for C3 compared to C2 for both the vertical and the horizontal STRF": It's hard to see evidence for that in the data.

Based on your next comment we reanalysed the kernels of C2 and C3. This resulted in a significant difference in peak timing between C2 and C3.

(6) When presenting kernels as in Figure 3d and Figure 4b, extend the time axis to positive times until the kernel goes to zero. This "prediction of future stimuli" allows the reader to see the degree of correlation within the stimulus, which affects how one interprets the shape of the kernel. Also, plotting the entire peak gives a better assessment of whether there are any shape differences between conditions. An alternative is to compute the kernel via deconvolution, which gets closer to the actual causal kernel, but that procedure tends to highlight high-frequency noise in the measurement.

We replotted the kernels in Figure 3d and 4b to show positive times. The kernels of C2 and C3 stayed at a positive level. Going back through the data we found a severe decrease in GCaMP signal in the first 2 seconds of the recording. We reanalyzed the kernels by ignoring the first seconds. All kernels now go back to zero. The shape of the kernels did not change but we now find a significant difference in peak timing between C2 and C3. Thank you for pointing this out.

(7) Line 280 "simultaneously blocked C2 and C3 using Kir2.1": First use of that acronym. Please explain what the method is.

We now explain “we simultaneously blocked C2 and C3 by overexpression of the inwardrectifying potassium channel Kir2.1”

(8) Line 350 "temporal dynamics for C2 silencing": suggests "dynamics of silencing"; maybe better "response dynamics during C2 silencing".

Edited as suggested

(9) Figure 7: Explain the details of the stimulus containing two subsequent on edges. What happens between one edge and the next? Does the screen switch back to black? Or does the second edge ride on top of the final level of the first edge? This matters for interpreting the response.

Yes, the screen turns dark between subsequent edge presentations. We added a sentence to the methods to clarify that.

(10) Line 402 "novel, critical components of motion computation.": This seems exaggerated. At the behavioral level, motion computation is mostly unaffected, except for some details of time resolution. Whether those matter for the fly's life is unclear.

We deleted the word ‘critical.’

(11) Line 413 "GABAergic inhibition required for motion detection is mediated by C2 and C3": Again, this seems exaggerated. Motion *detection* appears to work fine, but the *discrimination* of two closely successive motion stimuli is affected. The rest of the text does properly distinguish "discrimination" from "detection".

We changed the title to say: ‘GABAergic inhibition in motion detection is mediated by C2 and C3.’

(12) Line 489 "Whereas the role of C2 and C3 for the OFF pathway may be more generally to suppress neuronal activity,": Unclear to what this refers. The present report emphasizes that there is no effect on OFF activity (Figure 5).

We did not see an effect of T5 responses to OFF flashes as shown in Figure 5 but we found a significant reduction of DS when silencing C2, as well as slightly overall increased responses to all directions for C2 and C3 silencing, which was significant for null directions when silencing C2. This is shown in Figure 6.

Typos:(1) Line 521.

Fixed

(2) Line 1170: context of the citation unclear.

Fixed